# Machine Learning Protocols in Early Cancer Detection Based on Liquid Biopsy: A Survey

**DOI:** 10.3390/life11070638

**Published:** 2021-06-30

**Authors:** Linjing Liu, Xingjian Chen, Olutomilayo Olayemi Petinrin, Weitong Zhang, Saifur Rahaman, Zhi-Ri Tang, Ka-Chun Wong

**Affiliations:** 1Department of Computer Science, City University of Hong Kong, Hong Kong, China; jinglliu3-c@my.cityu.edu.hk (L.L.); xingjchen3-c@my.cityu.edu.h (X.C.); opetinrin2-c@my.cityu.edu.hk (O.O.P.); weitzhang6-c@my.cityu.edu.hk (W.Z.); srahaman2-c@my.cityu.edu.hk (S.R.); gerintang@163.com (Z.-R.T.); 2Hong Kong Institute for Data Science, City University of Hong Kong, Hong Kong, China

**Keywords:** machine learning, early cancer detection, liquid biopsy

## Abstract

With the advances of liquid biopsy technology, there is increasing evidence that body fluid such as blood, urine, and saliva could harbor the potential biomarkers associated with tumor origin. Traditional correlation analysis methods are no longer sufficient to capture the high-resolution complex relationships between biomarkers and cancer subtype heterogeneity. To address the challenge, researchers proposed machine learning techniques with liquid biopsy data to explore the essence of tumor origin together. In this survey, we review the machine learning protocols and provide corresponding code demos for the approaches mentioned. We discuss algorithmic principles and frameworks extensively developed to reveal cancer mechanisms and consider the future prospects in biomarker exploration and cancer diagnostics.

## 1. Introduction

When cells mutate, they could divide uncontrollably and eventually form cancer [1]. According to the World Health Organization, cancer accounts for nearly 10 million deaths in 2020. Unfortunately, this number is estimated to be still climbing in the following decades and will reach 27 million new cases in 2040 [2]. As the second factor of death, cancer accounts for one-sixth of deaths worldwide each year [3]. Therefore, fighting against cancer is a huge challenge for global public health. Early detection, followed by tailored site-specific treatment, plays an important role in the front-line cure of cancer and could reduce the eventual mortality of cancer patients [4,5,6].

Cancer is associated with mutated genes; and genetic analysis is increasingly applied in cancer diagnosis [7]. The traditional methods for genetic testing on cancer patients are sampling from tumor tissues. However, tumor tissue biopsy is limited by several drawbacks such as invasive acquisition, clinical complications, sample preservation, and tumor heterogeneity [8,9,10].

Liquid biopsy [7,11], which surmounts the limitation of tissue biopsy, is evaluated as a potential tool for early cancer detection and monitoring [12]. By sampling from blood, stool, urine, saliva, and other fluid samples, liquid biopsy provides a non-invasive and feasible cancer detection service [13,14,15,16]. Compared with tissue biopsy, liquid biopsy is also more comprehensive to evaluate tumor heterogeneity since tumor sites can release aberrant signals into body fluid [17,18]. Researchers paid significant attention to the different components from liquid biopsy which are associated with cancers [19,20,21,22,23].

As the possibility or severity of tumor in the body is relevant to the liquid biopsy components, accurate cancer prediction based on the characteristics of these components becomes a significant problem. The application of machine learning protocols has been widely studied in recent years, proving to be valuable in early cancer detection. Nevertheless, the required knowledge to implement these methods is high, posing an obstacle to researchers who are looking to get started on liquid biopsy analysis and early cancer detection. Therefore, this review not only focuses on the published research of machine learning in early cancer detection but also demonstrates the entire implementation procedure in this effort.

The rest of this review is written in the following sections. Section 2 introduces the procedures of implementing machine learning, including data preprocessing, model selection, model evaluation, and hypothesis testing. Section 3 summarizes the mainly liquid biopsy components associated with cancer. Section 4 is an overview of the most widely used machine learning algorithms and the relevant literature with corresponding datasets. Section 5 is the discussion on this topic. For all machine learning protocols and algorithms, we provided the code demo as a tutorial available at (https://github.com/ElaineLIU-920/Code-Deme-for-ML-procedures-and-algorithms, accessed on 10 June 2021).

## 2. Machine Learning Related Procedures

The data sets of cancer liquid biopsies are large and complex. Therefore, it is difficult to deal with using traditional methods. Machine learning algorithms, as a potential tool, can automatically analyze and identify regularities from data and then predict future data based on the obtained experience. For machine learning, the detection of cancer is regarded as a supervised problem, which is called a classification task. In this section, we focus on the supervised machine learning protocols and some of the preparatory work before implementing these methods. This section is organized according to a typical workflow for supervised machine learning. Firstly, we will discuss some techniques for data preprocessing. Moreover, model evaluation and selection methods which include the performance metrics for supervised learning are also discussed. Next, we will introduce the hypothesis test to indicate the statistical significance.

### 2.1. Data Preprocessing

Data preprocessing is a fundamental step of the machine learning implementation, which has been stated to have a significant influence on the performance of machine learning models [24,25]. Data preprocessing consists of missing-value solution, normalization, dimension reduction, and feature reconstruction. As the future data is unknown in reality, we suggest that all data preprocessing methods are only applied to training data.

#### 2.1.1. Missing Value

Missing value cannot be avoided in a dataset, which may create an obstacle for predictors. Inappropriately handling strategy will easily result in extracting poor knowledge, and wrongly prediction [26].

The first option to deal with this problem is to delete samples with missing values [27,28,29], which may result in discarding a large number of samples and increasing bias prediction [30]. Alternatively, the missing value can be filled by the mean, mode, or a random value [25]. Moreover, some model-based methods are also employed to predict the missing value [30]. Model-based methods do neither delete missing-value samples nor fill the value by simple imputation; Instead, it builds a model for the missing feature based on inferences from existing complete data.

Model-based methods consist of two steps: (1) Build a regression or classification model based on complete samples for the feature which is corresponding to the missing values; (2) Predict on the incomplete samples with its existed feature as input, and then the output is an estimate of missing value [31].

#### 2.1.2. Normalization

The main advantage of implementing normalization is that it prevents the predictions of later stages from being dominated by relatively large or small values in the data set. Besides, normalization is significant to ensure comparability over different samples. In this section, we will introduce three commonly used normalization methods, namely Z-Score standardization, Max-min normalization, and Decimal scaling [32].

Z-Score standardization. In Formula (Equation 1), *A* is feature (attribute), $x_{A}$ is the original value of feature *A*, $x_{A}^{\prime }$ is the normalized value; μ is the mean of feature *A*, and σ is the standard deviation of feature *A*.
(1)$$x_{A}^{\prime }=\frac{x_{A}-\mu }{\sigma }$$Max-Min normalization. Max-Min normalization, also called deviation standardization, is transformed by Formula (Equation 2), where min is the minimum of feature *A*; max is the maximum of the feature *A*.
(2)$$x_{A}^{\prime }=\frac{x_{A}-\min}{\max-\min}$$Decimal scaling. This method is realized by moving the decimal point position according to the absolute maximum of feature *A*. In Formula (Equation 3), *j* is the smallest integer such that all $x_{A}^{\prime }$ is less that 1, $j=\lceil \log_{10}\max\rceil$. Here, max is the maximum of the feature *A*.
(3)$$x_{A}^{\prime }=\frac{x_{A}}{10^{j}}$$

#### 2.1.3. Dimension Reduction

Feature is the observation of samples, which is also synonymous with input variables or attributes. The dimension of the dataset is the number of variables measured on each sample, equal to the number of features. Owing to the development of detection technology, the available samples have increased explosively in terms of dimension. When machine learning algorithms are applied to these high-dimensional data [33,34], dimension curse becomes a crucial issue to resolve, which is especially severe in bioinformatics [35,36].

One of the problems with the high-dimensional dataset is that some algorithms tend to perform poorly on high-dimensional data, as not all features are valuable for prediction. In many cases, a large amount of the features are irrelevant or redundant with the learning task, resulting in overfitting for learning models [32]. In addition, high dimensional data will also increase computation time as well as the memory of storage. Moreover, if the dimension of data is very high, visualization becomes quite difficult.

Feature extraction (also known as feature transformation, feature projection or dimension reduction specifically) and feature selection are two dimension reduction techniques [37] to solve these problems. The choice of feature extraction or feature selection depends on different data types and applications. We will next briefly introduce some typical approaches for dimension reduction.

Feature Extraction

Feature extraction method develops a transformation from the original high-dimensional feature space into a new low-dimensional space. The essence of feature extraction reduction is to learn a mapping function $f:X\to X^{\prime }$, where *X* is the original data, and $X^{\prime }$ is a low-dimensional vector representation after data mapping. Linear mapping and non-linear mapping methods are two main types to implement feature extraction [38]. Linear mapping is mainly represented by principal component analysis (PCA) [39,40], linear discriminant analysis (LDA) [41], and non-negative matrix factorization (NMF) [42], while non-linear mapping is mainly represented by locally linear embedding (LLE) [43] and Isomap [44].

The advantage of feature extraction is that it decreases the dimension of feature through data transformation, which enables obtaining a lower feature space without losing information. However, it is precisely for this reason that the new space is obtained from the linear or non-linear transformation of the original space, causing the inexplicability of the new features.

B.Feature Selection

Different from feature extraction, feature selection directly selects a valuable subset features and removes noisy, redundant, or irrelevant features from the original dataset, which only contains the important information to solve the problem [45,46,47]. Based on different pathways of combining feature selection strategy with machine learning models, feature selection techniques are categorized into three types: filter method, wrapper method and embedded method [48].

Filter methods, independent of any learning models, assess the importance of features based on the statistical and intrinsic properties of the original dataset. In this setup, importance ranking is adopted as the principal criteria for feature selection. By reserving high-scoring features and removing low-scoring features, a subset with a lower dimension of features is obtained. Many filter-type methods have been studied, including Pearson correlation coefficient [49], F-statistic [50], Chi-squared-statistic [51] and Mutual information [52].

Wrapper methods adopt different search algorithms to generate the subsets of features. Subsequently, a specific subset is evaluated by training and testing the performance of the classification model, which is wrapped in the search algorithm. The whole process works iteratively until the highest learning performance is achieved or the desired number of selected features is obtained. A wide range of search strategies can be used, including Sequential Selection Algorithms, Recursive Feature Elimination, and Meta-heuristic Algorithms (e.g., genetic algorithm) [53,54].

Embedded methods explore the optimal subset of features during the process of constructing a learning model. Similar to the wrapper methods, the embedded methods are specific to the adopted machine learning algorithm. Least absolute shrinkage and selection operator, Elastic net and Ridge regression are three typical regularization algorithms [55,56].

Detail comparison of these three pathways to implement feature selection is discussed in [48,57]. As feature selection merely explores a valuable subset of the original feature, it retains the semantics of the original features, which possesses the advantage of interpretable analysis. However, some information may be lost when employing feature selection methods, as only a subset is reserved, and some of the features will be omitted.

Feature extraction, as well as feature selection, has the ability to improve model performance, computational efficiency, utilization of memory storage, and data visualization. Therefore, both of these two methods are employed as effective dimension reduction techniques, used alone or in combination.

#### 2.1.4. Feature Construction

Feature construction is also known as attribute generation. Different from dimension reduction, in some cases, the features may be insufficient to describe the problem for learning models. Therefore, feature construction is adopted utilized to enrich the data. According to the definition, taken from Motoda and Liu [58], feature construction aims to discover the hidden relationships of original features by constructing new high-level features. Similar to feature selection, the process of constructing feature can also be categorised into three classes: filter methods, wrapper methods, and embedded methods [59,60]. For numerical features, simple algebraic operators such as addition, subtraction, multiplication, and division are often used to compound features.

### 2.2. Model Evaluation

Model evaluation is the process of assessing the performance of models on the future data [61]. In the straight forward, it aims to evaluate how well the built model by estimating the generalization error on unseen data. A good machine learning model should perform well not only on the training data but also on the future data. Therefore, before implementing a model for production, we should be fairly sure that the performance of the model will not decline when confronted with the new data. For most practical applications, the true performance of the model cannot be calculated as we do not have real future data. Hence, it is important to use new data for model evaluation to prevent the likelihood of overfitting problems to the training set. Holdout, bootstrap, and cross-validation are most commonly used method for model evaluation [62,63,64].

#### 2.2.1. Holdout Method

Holdout method is the simplest model evaluation method, which directly splits the dataset into two portions: training set and test set. For example, we randomly choose 2/3 of the whole dataset as the training set and 1/3 as the test set. Firstly, we utilize the training set to fit and build the model. Subsequently, we evaluate the built model on the test set by comparing the predictions of the label and the ground truth. To some extent, the test set represents the new and unseen data in practice. As the estimation result obtained by applying the holdout method once is often not reliable, it necessitates the repeating of splitting and evaluating several times, which is called the repeated holdout method. The average performance evaluation is reported as the final estimation result. We usually utilize about 2/3 to 4/5 of the dataset for training and the rest for testing.

It should be noted that we cannot train and evaluate the model based on the training dataset simultaneously, which is called resubstitution evaluation or resubstitution validation. As resubstitution evaluation would introduce optimistic bias due to overfitting on resubstitution samples, we cannot ascertain whether the model works because it remembers the training data or because it could generalize well on new data.

#### 2.2.2. Cross-Validation

The basic idea of cross-validation is to divide the data into different subsets. In this setup, some of these subsets are used to train the model and the rest are used to test the model until all the samples have been used for testing. k-fold cross-validation strategy is most commonly used in the classification research [65]. With *k*-fold cross-validation, the dataset is partitioned into *k* disjoint subsets, the union of which is equivalent to the whole dataset. A single subset from these *k* disjoint subsets is retained as the test data to evaluate the classifier, and the remaining k−1 subsets are used as training data. This process is then repeated for *k* times until all subsets are used as the test data exactly once. The performance evaluation results on *k* test set are averaged as the performance estimation for the classifier.

The step-by-step instruction of *k*-fold cross-validation is summarized as below. Figure 1 is the diagram of *k*-fold cross-validation.

**Step 1:** Randomly split the original dataset into *k* equal folds.**Step 2:** Select one of these folds as test set, and the remaining k−1 folds as training set to build model.**Step 3:** Compute generalization performance of the built model on the test set.**Step 4:** Repeat step 2 to step 3 for *k* times until each fold has and only has one chance to act as the test set, and the remaining folds act as the training set.**Step 5:** Report the average of generalization performance on all test sets as an estimations of the model performance.

The different values of *k*, which is usually five, ten or equal to the number of instances in the dataset, determine the different subtypes of cross-validation. Assuming that the dataset includes *n* samples, if k=n, we obtain a special case of the cross-validation, namely, the leave-one-out cross-validation (LOOCV). Obviously, the LOOCV method is not affected by the partition of samples, as there is only one unique way for *n* samples to be divided into *n* subsets, each of which contains only one sample. Although the evaluation results of the LOOCV method are often considered to be more accurate, LOOCV method also has unbearable computational overhead when the dataset is relatively large. For example, LOOCV needs to build 1 thousand models if the dataset contains 1 thousand samples. However, 5-fold cross-validation and 10-fold cross-validation only need to build five and ten models, respectively.

#### 2.2.3. Bootstrapping

The bootstrapping method is a re-sampling technique to draw sample data repeatedly with replacement from the original dataset, proposed by Bradley Efron in 1979 [66]. The workflow of bootstrap method is summarized as following:**Step 1:** The size of original dataset is *n*. We randomly select one instance from this dataset and then assign it to the $j_{th}$ bootstrap dataset. Repeating this process until the size of $j_{th}$ bootstrap sample reaches *n*.**Step 2:** Fit a model to $j_{th}$ bootstrap dataset and compute the performance.**Step 3:** Repeat Step 2 and 3 for *b* times. Calculate the model performance as the average over the *b* estimates. If accuracy is the performance metric, then the model performance for bootstrapping is:(4)$$ACC_{boot}=\frac{1}{b}\Sigma_{j=1}^{b}\frac{1}{n}\Sigma_{i=1}^{n}\left(1-L({\hat{y}}_{i},y_{i})\right)$$

In 1983, Bradley Efron described the 0.632 Estimate [67] to address the bias of the bootstrap approach aforementioned. The bias in the conventional bootstrap method is owning to the fact that the bootstrap approach only utilize approximately 63.2% of the samples from the whole dataset. For example, we can calculate the probability that a specific sample, from a dataset with size *n*, is not selected as as following:(5)$$P_{N}={\left(1-\frac{1}{n}\right)}^{n}$$

The value of Equation (Equation 5) is asymptotically equivalent to $\frac{1}{e}\approx 0.368$ when n→∞. Therefore, the probability that the specific sample is chosen as:(6)$$P_{C}=1-{\left(1-\frac{1}{n}\right)}^{n}\approx 0.632$$

Subsequently, to adjust the bias that is owing to the sampling strategy, Bradley Efron introduced the 0.632 Estimation method, computed by the Formula (Equation 7):(7)$$ACC_{0.632boot}=\frac{1}{b}\Sigma_{i=1}^{b}\left(0.632ACC_{h,i}+0.368ACC_{r,i}\right)$$
where $ACC_{r,j}$ is the resubstitution accuracy, and $ACC_{h,j}$ is the accuracy on out-of-bag samples (samples which are not selected as the bootstrap samples). The 0.632 Boostrap could address the pessimistic bias, however, an optimistic bias may occur. Therefore, 0.632 + Bootstrap was proposed [68].
(8)$$ACC_{0.632+boot}=\frac{1}{b}\Sigma_{j=1}^{b}\left(\omega ACC_{h,j}+\left(1-\omega \right)ACC_{r,j}\right)$$

Instead of using a fixed weight ω=0.632, 0.632 + Bootstrap compute the weight ω as
(9)$$\omega =\frac{0.632}{1-0.368R}$$
where *R* is the relative overfitting rate:(10)$$R=\frac{\left(-1\right)\left(ACC_{h,j}-ACC_{r,j}\right)}{\gamma -(1-ACC_{h,j})}$$
where γ is the no-information rate. We can calculate γ through fitting a model to a dataset that contains all possible combinations between features $x_{i}^{\prime }$ and target class labels $y_{i}$:(11)$$\gamma =\frac{1}{n^{2}}\Sigma_{i=1}^{n}\Sigma_{i^{\prime }=1}^{n}L\left(y_{i},f\left(x_{i^{\prime }}^{\prime }\right)\right)$$

Additionally, the no-information rate γ could be estimates as:(12)$$\gamma =\Sigma_{k=1}^{K}p_{k}\left(1-q_{k}\right)$$
where $p_{k}$ is the percentage of examples belonging to class *k* and observed in the dataset, and $q_{k}$ is the percentage of examples that the classifier predicts to belong class *k*.

#### 2.2.4. Performance Evaluation Metrics

There are four types of possible outcomes for classification tasks, true positive, true negative, false positive, and false negative. The definition of these four terms is listed in Table 1.

These four outcomes are often listed on the confusion matrix. The following confusion matrix (Table 2) is an illustration for the case of binary classification.

Next, we will introduce some model evaluation metrics.

Accuracy (also known as recognition rate) is defined as the fraction of correct predictions. It can be calculated easily by dividing the number of correct predictions by the number of total predictions.
(13)$$Accuracy=\frac{TP+TN}{TP+TN+FP+FN}$$

Precision (also known as positive predictive value, PPV) is defined as the fraction of correct positive predictions among all of positive predictions.
(14)$$Precision=\frac{TP}{TP+FP}$$

Recall (also known as sensitivity, true positive rate, TPR) is defined as the ratio of true positive predictions with respect to all of the examples that truly belong in positive class.
(15)$$Recall=\frac{TP}{TP+FN}$$

$F_{\beta }$ score consider both precision and recall together as an evaluation index. The β parameter allows us to control the trade-off of importance between precision and recall. β<1 focuses more on precision while β>1 focuses more on recall. When β=1, it is called $F_{1}$ score.
(16)$$F_{\beta }=\frac{(1+\beta^{2})\times Precision\times Recall}{\beta^{2}\times Precision+Recall}$$

Brier score is used to check the goodness of a predicted probability score, whose values range between 0 and 1. For binary classification, the score is given by:(17)$$BS=\frac{1}{n}\Sigma_{i=1}^{n}{\left(p_{i}-o_{i}\right)}^{2}$$
where $p_{i}$ is the prediction probability, and the term $o_{i}$ is equal to 1 if the event occurred and 0 if not.

Receiver Operating Characteristic Curve (ROC Curve) is the plot between the true positive rate and false positive rate. Following (Figure 2) is an example of the ROC curve. The area under the ROC curve (AUC) is to measure how well the classifiers make correct predictions on the different thresholds.

### 2.3. Model Selection

With the development of machine learning, researchers proposed many efficient machine learning algorithms. For each algorithm, there are several hyperparameters that can be tuned to fit different datasets. Using different hyperparameters and algorithms to fit the training data sets results in different candidate models. As we are usually interested in obtaining the best-performing model from these candidate models, we need to find an approach to evaluate their respective performance in order to rank them. Model selection is the process of selecting the best machine learning model from the candidate models, which are built based on the training dataset. It involves the selection of different types of models (e.g., KNN, SVM, RF, etc.) and the selection of models with different hyperparameters for a certain type (e.g., different kernels for SVM).

As mentioned before, it is essential to evaluate our model with new data to prevent the possibility of overfitting on the training set. However, in order to select the best model, we need to evaluate the candidate models while building the model. In light of the fact that we cannot evaluate the candidate models on the test set. Otherwise, we will obtain a model that performs best on the test set but may not generalize well in practice. To evaluate the model as we build and adjust the model, we create a third subset of the dataset, called the validation set. If we have plenty of data, which may be at least 1000 to infinite, we could straightforward create the validation set. To evaluate the model as we build and tune the model, we could randomly split the full dataset into training, validation, and test sets. Then, we would fit candidate models on the training set with a different configuration of hyperparameters and algorithms. Subsequently, we can evaluate the performance of candidate models on the validation set and select the winning model which performs best (model evaluation and selection). With the hyperparameters of the best model, we retrain it using the training + validation set, and the generalization performance of the final model is evaluated on the test set (model evaluation). If the performance on the test set is similar to the performance on the validation set, there is reason to believe that the model will perform well on future data. Finally, we retrain the model on the full dataset (training, validation, and test set) for production use.

However, we rarely have such sufficient datasets in practice. We mainly have two approaches, re-sample methods and analytical methods, to implement model selection for a limited size of the dataset [69].

#### 2.3.1. Re-Sample Methods

For re-sample methods, we expand the sample size by repeating a random re-sampling training set and then compute the average of prediction error as the estimation. In general, we split the training dataset into sub-training and validation sets. Sub-training set is used to fit candidate models for different algorithms and hyperparameters. The validation set is used to evaluate these candidate models and select the best model. Model evaluation does not change, in which test set is still utilized to estimate the performance of the final selected model.

We can adopt the aforementioned methods (holdout, bootstrapping and cross-validation) of model evaluation to split the training dataset again. By far, the most widely used is the cross-validation method, which includes many subtypes. Here, nested cross-validation method [70] will be detail for an example. Up to now, we have two tasks: the first task is to select the best model across candidate algorithms and corresponding hyperparameters; and the second task is to estimate the generalized performance of the best model. The nested cross-validation method includes an inner loop and an outer loop. In the inner loop, the target is to select the best model, whereas, in the outer loop, the target is to estimate the generalization performance of the best model selected by the inner loop. Figure 3 illustrates the procedure of the nested cross-validation. It works as follows:**Step 1:** Randomly split the whole dataset into *K* equal folds (outer loop).**Step 2:** Select one of them as the test set, and the remaining k−1 folds as the training set.**Step 3:** Randomly split the training set into $K^{\prime }$ equal sub-folds (inner loop).**Step 4:** Select one of the sub-folds as the validation set and the remaining $k^{\prime }-1$ folds as the sub-training set. Then we train candidate models under different algorithms and hyperparameters with the sub-training set. Next, we evaluate the performance of candidate models on the current validation set.**Step 5:** Repeat step 4 for $k^{\prime }$ times, so that each sub-fold has and only has one chance to act as the validation set, and the remaining sub-folds act as the sub-training set.**Step 6:** We then compute the average performance of candidate models on all validation sets and select the winning model with the best performance.**Step 7:** With the hyperparameters of the best model from Step 6, we retrain it with the whole training set and then evaluate the generalization performance of the best model on the current test set.**Step 8:** Repeat step 2 to step 6 for *k* times, so that each fold has and only has one chance to act as the test set, and the remaining folds act as the training set.**Step 9:** Report the average of generalization performance on all test sets as an estimate of the model performance.

Lastly, we retrain the best model using the whole dataset for deployment. For brevity, nested CV with *K* outer folds and $K^{\prime }$ inner folds is denoted as $K\times K^{\prime }$ nested CV. Typical values for $K\times K^{\prime }$ are 10×10, 10×5, or 5×5, etc.

#### 2.3.2. Analytical Measures

Compared with re-sample methods, the analytical methods not only evaluate model performance but also consider the model complexity. In addition, as analytical methods approximate the test error from the training error, which does not need to repeat several times, it could improve the efficiency of model selection. In this part, three typical used analytical criteria are introduced for model selection.

Akaike Information Criterion (AIC) is a scoring criterion to measure the performance of statistical models, named for the Japanese statistician Hiroji Akaike who proposed AIC in 1973 [71].
(18)AIC=−2·loglike+2·d

Formula (Equation 18) is a mathematical formulation of AIC, where loglike is the maximized log-likelihood; *d* is a measure of model complexity, such as the number of parameters for linear models. It is noted that the form of *d* for nonlinear and complex models differ and should be carefully derived. To use AIC for model selection, we simply choose the model with the smallest AIC over the set of models considered.

Bayesian Information Criterion (BIC), also known as the Schwarz criterion, was derived from Bayesian probability, and inferenced by Schwarz [72]. Like AIC, it is applicable for models that are fitted by the maximum of likelihood. If we use the same formalism defined in Formula (Equation 18), the generic form of Bayesian Information Criterion is defined as follows:(19)BIC=−2·loglike+(logn)·d

It is straightforward to find that BIC is proportional to AIC. Compared with AIC, BIC punishes heavily on models, which possess more parameters and higher complexity. Although it looks similar, the original idea of BIC is not similar to AIC, but obtained from a Bayesian perspective.

Minimum Description Length (MDL) is motivated from an optimal coding viewpoint, proposed by Rissanen [73]. MDL recommends us to select the model from an information theory perspective. If we want to transmit our model and the prediction, a good solution from the view of coding is to encode the message with shortest length. According to Shannon’s theorem [74], the length to describe our problem is:(20)Length=−logPr(y|θ,M,X)−logPr(θ|M)

In Formula (Equation 20), *M* is our model with θ as parameter. Pr(y|θ,M,X) is the conditional probability of the model output with attribute X. The first term of Formula (Equation 20) represents the average code length for transmitting the difference between the output of the model and the ground truth, whereas the second term represents the average code length for transmitting the model parameter vector θ.

One advantage of the analytical measure for the model selection approach is that it does not require a validation dataset. It means that all of the data can be used to build the model, and we can score the candidate models directly. However, the analytical measure also has the limitation of the inability to form general statistics across different types of models. For a more detailed discussion about analytical measures, the material can be obtained from [75].

#### 2.3.3. Hyperparameter Tuning

The hyperparameters of machine learning algorithms enable the model to be tailored to different datasets. Therefore, hyperparameter tuning, which refers to the searching of an appropriate hyperparameter configuration, is an important process for the application of machine learning. Grid search, random search, Bayesian optimization, and meta-heuristic algorithms are most commonly used for hyperparameter tuning.

Grid search is an exhaustive search strategy exploring a grid of evenly spaced values. Generally, grid search can find the global optimum value by setting a large search range and a fine grid. It involves generating a uniform grid of hyperparameter configuration across the search space. With this search strategy, we simply build the model for each potential combination of all of the hyperparameters and evaluate the model to select the one which achieves the best results. The downside is that the number of potential hyperparameter combinations to be explored grows exponentially with the number of hyperparameters. It is quite inefficient to try all hyperparameter combinations one by one, which could take days or even weeks, especially on a large dataset.

Different from grid search, random search simply draws some random samples instead of trying all hyperparameter settings. This strategy randomly samples model hyperparameters following a sampling distribution (e.g., uniform) for a number of iterations. For each iteration, we build the model under a hyperparameter combination, which is randomly sampled from the aforementioned distribution. Subsequently, we evaluate each chosen hyperparameter configuration and select the best one. On account of randomness, it is not guaranteed that random search always finds the optimal solution.

Meta-heuristic algorithm is a generic optimization framework that can resolve almost all optimization problems as it is a problem independent. The iterative generation process of meta-heuristic algorithm realizes the robust searching mechanism by balancing exploration (diversification) and exploitation (intensification) under different intelligent concepts. Therefore, it enables the black-box optimization problem of hyperparameter tuning solvable with an optimal or near-optimal solution. Genetic algorithm [76], Particle Swarm optimization [77], Simulated Annealing [78], and Tabu Search [79] are already introduced for hyperparameter tuning.

Bayesian optimization for machine learning parameter tuning was proposed by J. Snoek (2012) [80]. It works under the assumption that the mapping between hyperparameter setting and generalization performance was sampled from a Gaussian process. We first construct the distribution by the observation of hyperparameters and corresponding generalization performance. Subsequently, the acquisition function was adopted to determine the next point of hyperparameters to evaluate the performance and add the observation to update the distribution. We iteratively repeat these two steps until converging to an optimum. In this setup, the information of the previous hyperparameter setting is included to adjust the exploring process.

### 2.4. Hypothesis Testing

Once we obtain the final model, we usually want to compare our method with the state-of-the-art methods to prove that it beats or performs as well as the advanced method. With model evaluation methods and performance metrics, it seems possible to compare the performance of the different models by first using an evaluation method to measure certain performance metrics of the models and then comparing the value of performance directly. However, the performance of models and the difference between models may be misleading because of the sampling error instead of essential differences. To make the performance of the model statistically significant, we will introduce hypothesis testing in this part.

Hypothesis testing is a statistical inference method to distinguish whether the results are due to sampling error or intrinsic differences. For hypothesis testing, we firstly compute a statistic from the samples and assume that it follows a certain distribution. If the probability of the statistic to obey this distribution is very low, we may reject the hypothesis; If not, we may accept it. For example, if we have a model with an average error rate which is $\epsilon_{0}$. In hypothesis testing, we may assume that the error rate is less than $\epsilon_{0}$. If the test result is consistent with the hypothesis, we accept the hypothesis; otherwise, we should reject the hypothesis as the error rate has a high probability to be greater than $\epsilon_{0}$.

Let us take the error rate under Student’s *t*-test [81] for example. If we adopt *k*-fold cross-validation, the model evaluation process provides *k* error rates as we split the dataset into *k* folds. We denote these error rates with ${\hat{\epsilon }}_{1},{\hat{\epsilon }}_{2},\ldots ,{\hat{\epsilon }}_{k}$, then the average error rate and standard deviation are μ (Formula (Equation 21)) and σ (Formula (Equation 22)).
(21)$$\mu =\frac{1}{k}\Sigma_{i=1}^{k}{\hat{\epsilon }}_{i}$$
(22)$$\sigma =\sqrt{\frac{1}{k-1}\Sigma_{i=1}^{k}{\left({\hat{\epsilon }}_{i}-\mu \right)}^{2}}$$

We can compute the *t*-statistic value following Formula (Equation 23), which obeys a *t*-distribution of *k* − 1 degree of freedom.
(23)$$t_{0}=\frac{\sqrt{k}\left(\mu -\epsilon_{0}\right)}{\sigma }$$

For $H_{0}:\mu \leq \epsilon_{0}\;vs.\;H_{1}:\mu \geq \epsilon_{0}$, we fail to reject the hypothesis with level of significance α if $t_{0}\in \left(-\infty ,t_{\alpha ,k-1}\right)$ or *p*-value is greater than the α.

#### 2.4.1. Paired *t*-Test

Paired *t*-test [82] is a specific Student’s *t*-test, which is used when the two samples are matched or paired. It is a two-sided test for the null hypothesis that two samples have identical average values.

When comparing machine learning models using paired *t*-test, we firstly evaluate each model on the same *k*-fold cross-validation split of the dataset and compute a performance score for each split. For example, we use $\epsilon_{1}^{A},\epsilon_{2}^{A},\ldots ,\epsilon_{k}^{A},$ and $\epsilon_{1}^{B},\epsilon_{2}^{B},\ldots ,\epsilon_{k}^{B},$ to denote the test error rate of model A and model B in the same split. Secondly, we calculate the difference of each pair $d_{i}=\epsilon_{i}^{A}-\epsilon_{i}^{B}$. If these two models perform the same, the mean of their difference should be zero. Therefore, we can compute the mean μ and the variance $\sigma^{2}$ of $d_{1},d_{2},\ldots ,d_{k}$. Next, we compute the *t*-statistic value by Formula (Equation 24).
(24)$$t_{0}=\left|\frac{\sqrt{k}\mu }{\sigma }\right|$$

For $H_{0}$: Model A and Model B perform the same vs. $H_{1}$: Model A and Model perform differently, we fail to reject the hypothesis with the level of significance α if $t_{0}$ is less than $t_{\alpha /2,k-1}$ or *p*-value is greater than the α. It means that there is no significant difference between model A and model B.

Valid use of paired *t*-test is based on the independence of each evaluation. However, in this case, the sub-training sets have overlap with each other, which means they lack independence in the evaluation. If we stick with this hypothesis test, it will lead to overestimating the probability of rejecting the hypothesis.

#### 2.4.2. 5 × 2 Cross-Validation Paired *t*-Test

To address this problem, 5 × 2 cross-validation, which repeats the 2-fold cross validation five times, was adopted as the evaluation method. We randomly scramble the dataset before each 2 fold cross validation to ensure that each observation occurs only once in the training or test dataset. Then, the evaluation results are tested using a paired *t*-test. For 5 × 2 cross-validation paired *t*-test [82], the computation of statistic value is slightly different with paired *t*-test. If we define $d_{i}^{j}\left(i=1,2;j=1,2,\ldots ,5\right)$ to denote the difference of test error rate between model A and model B in the *i* fold of *j* repetition, then the average performance in each repetition is $\mu_{j}=\left(d_{1}^{j}+d_{2}^{j}\right)/2$ and the variance is $\sigma_{j}^{2}={\left(d_{1}^{j}-\mu_{j}\right)}^{2}+{\left(d_{2}^{j}-\mu_{j}\right)}^{2}$. With these denotation, we define the statistic of 5 × 2 cross-validation paired *t*-test in Formula (Equation 25).
(25)$$t_{0}=\left|\frac{\mu_{1}}{\sqrt{\frac{1}{5}\Sigma_{j=1}^{5}\sigma_{j}^{2}}}\right|$$

The statistic obeys a T-distribution of 5 degrees of freedom. Similar with paired *t*-test, if the statistic $t_{0}$ is less than $t_{\alpha /2,5}$, model A and Model B perform equivalently. Conversely, the performance of the two models is significantly different, and the one with a lower average error rate performs better.

#### 2.4.3. Wilcoxon Signed-Rank Test

Wilcoxon signed-rank test [83] is a non-parametric hypothesis test, proposed by Frank Wilcoxon in 1945. It applies to the case of two related or paired samples to assess whether their populations have the same distribution. In the specific, it checks whether the difference between the paired observations comes from a population with a median of zero. We can utilize Wilcoxon signed-rank test to compare two different models A and B, at the statistic level as following steps. It should be noted that the denotations not mentioned in this section are the same as in paired *t*-test.

**Setp 1: Build the null hypothesis.**$H_{0}$: the performance distributions of model A and model B are equal, $H_{1}$: the performance distributions of model A and model B are not equal.**Setp 2: Calculate the difference.** For i=1,2,…,k, calculate the difference of each pair $d_{i}=\epsilon_{i}^{A}-\epsilon_{i}^{B}$.**Setp 3: Rank the difference.** Order the difference according to its absolute value $|d_{i}|$ from the smallest to largest value. Define $r(|d_{i}|)$ to denote the rank. The rank of smallest $|d_{i}|$ is 1; Ties (pairs with equal $|d_{i}|$) ranks equal to the average of the orders they cross; The differences equal to zero are omitted when ranking. For example, if we have six difference values ($d_{1}=5$, $d_{2}=0$, $d_{3}=-2$, $d_{4}=5$, $d_{5}=1$, $d_{6}=6$), the absolute values of them are $|d_{1}|=5$, $|d_{3}|=2$, $|d_{4}|=5$, $|d_{5}|=1$, $|d_{6}|=6$. Therefore, the ranks of them are $r(|d_{1}|)=3.5$, $r(|d_{3}|)=2$, $r(|d_{4}|)=3.5$, $r(|d_{5}|)=1$, $r(|d_{6}|)=5$.**Setp 4: Compute the statistic.**$W^{+}$ is the sum of the ranks of the originally positive differences. Conversely, $W^{-}$ is for negative differences. The test statistic of Wilcoxon signed-rank test is $W=min(W^{+},W^{-})$. *W* can be compared to the critical value table for the Wilcoxon signed-ranks test in [84]. Let $k^{\prime }$ denote the number of pairs included in the ranking. If $W>W_{critical,k^{\prime }}$, we reject the null hypothesis. For the instance of step 3, the value of $W^{+}$ is 13, and the value of $W^{-}$ is 2. Therefore, *W* is 2. As $W_{critical,5}$ is 0 and $W>W_{critical,5}$, we reject $H_{0}$ and accept $H_{1}$. It means the model with a higher performance score is statistically significant. Moreover, the sum of the positive difference ranks ($W^{+}$ = 13) is larger than the sum of the negative difference ranks ($W^{-}$ = 2), showing a positive advantage from model A. Consequently, our analysis provides significant evidence that model A performs better than model B in statistical significance.**Setp 5: Compute the z-score.** If $k^{\prime }>30$, we can implement a large sample approximation. For the population of statistic *W*, the mean is $\mu_{W}$ (Formula (Equation 26)) and the standard deviation is $S_{W}$ (Formula (Equation 27)) [84].
(26)$$\mu_{W}=\frac{k^{\prime }\left(k^{\prime }+1\right)}{4}$$
(27)$$S_{W}=\sqrt{\frac{k^{\prime }\left(k^{\prime }+1\right)\left(2k^{\prime }+1\right)}{24}}$$Therefore, the *z*-score is
(28)$$z_{0}=\frac{W-\mu_{W}}{S_{W}}$$Then, we compare the obtained $z_{0}$ value to the critical value of normal distribution or calculate the *p*-value. As a general rule, we set the level of risk to be α=0.05. If *p*-value is less than 0.05 or the absolute value of $z_{0}$ is greater than $z_{\alpha /2}$, we will reject the null hypothesis.

#### 2.4.4. McNemar’s Test

For models that are both very large and built for large datasets, it usually takes several days or weeks to train a single model. Therefore, it is impractical or expensive to perform multiple copies of the model. McNemar’s Test [82,85,86] is capable of comparing the models that can be executed only once. It is named for Quinn McNemar, who proposed it in 1947 [85].

To implement McNemar’s Test, we adopt the holdout method to train and test models *A* and *B*. For each model, we record the classification results on the test set and tabulate the outcomes on the following Table 3. It is the contingency table which lists the detail of misclassification by model *A* and *B*. For example, $n_{00}$ is the number of samples misclassified by both model *A* and *B*; $n_{01}$ is the number of samples misclassified by model *A* but not by model *B*.

Similar to before, we have the null hypothesis that the two models have no difference in performance. In other words, the error rates of these two models are the same, which means that $n_{01}=n_{10}$. Consequently, we build the statistic as Formula (Equation 29). The statistic follows the $\chi^{2}$ distribution with 1 degree of freedom. At the significance level of 0.05, the critical value of 1 degree of freedom is 3.841 [87].
(29)$$\chi_{0}^{2}=\frac{{\left(|n_{01}-n_{10}|-1\right)}^{2}}{n_{01}+n_{10}}$$

Then, we compare the obtained value of $\chi_{0}^{2}$ to the critical value ($\chi_{1-\alpha /2}^{2}$ and $\chi_{\alpha /2}^{2}$) of chi-square distribution or calculate the *p*-value. For the classical approach, if $\chi_{0}^{2}<\chi_{1-\alpha /2}^{2}$ or $\chi_{0}^{2}>\chi_{\alpha /2}^{2}$, we should reject the null hypothesis, which assumes that model *A* and *B* performs equally. For the *p*-value approach, if *p*-value <α, we should reject the null hypothesis and state the conclusion that the performance of model *A* and *B* are significantly different. In this way, there is sufficient evidence at the significance level of α=0.05 to conclude that these two models perform at a different error rate level.

#### 2.4.5. Friedman Test and Post-Hoc Test

If one model performs well on some data sets and poorly on others compared to other models, how can we tell if this model outperforms the others or not? Friedman test and the corresponding post hoc test are employed to explore the answer to this question.

The Friedman test is a non-parametric hypothesis test to compare multiple models on different datasets, which is proposed by Milton Friedman [88,89]. The procedure of Friedman test involves ranking the models for each dataset separately and calculate the Friedman statistic to infer whether these models perform differently. Suppose we compare *k* models on *N* datasets and let $R_{i}$ represents the average rank of *i*th algorithm on all datasets. For example, if we have three models, *A*, *B*, and *C*, and four datasets, the best performing algorithm ranks 1, the second-best ranks 2 and the last one ranks 3. In the case of ties, the average of the ranks they across are assigned. Table 4 is an example of the ranking results, where $R_{1}=1.125$, $R_{2}=2.250$, $R_{3}=2.625$.

The null hypothesis states that there is no difference among all models, which means that the average rank $R_{i}$ should be equivalent. The average and variance of $R_{i}$ are $\left(k+1\right)/2$ and $\left(k^{2}-1\right)/12$, respectively. The variable $\chi_{0}^{2}$, defined by Formula (Equation 30), is distributed according to the $\chi^{2}$ distribution with k−1 degree of freedom when *k* and *N* are big enough.
(30)$$\begin{array}{cc} \chi_{0}^{2} & =\frac{k-1}{k}\cdot \frac{12N}{k^{2}-1}\Sigma_{i=1}^{k}{\left(R_{i}-\frac{k+1}{2}\right)}^{2}\\ \; & =\frac{12N}{k\cdot \left(k+1\right)}\left(\Sigma_{i=1}^{k}{R_{i}}^{2}-\frac{k{\left(k+1\right)}^{2}}{4}\right) \end{array}$$

However, the above statistic is over conservative and a new statistic $F_{0}$ as Formula (Equation 31) is adopted.
(31)$$F_{0}=\frac{\left(N-1\right)\chi_{0}^{2}}{N\left(k-1\right)-\chi_{0}^{2}}$$

$F_{0}$ follows a *F*-distribution with k−1 and (k−1)(N−1) degrees of freedom. Next, we compare the obtained value of $F_{0}$ to the critical value $F_{\alpha }\left(k-1,(k-1)(N-1)\right)$ or calculate the *p*-value. For classical approach, if $F_{0}<F_{1-\alpha /2}$ or $F_{0}>F_{\alpha /2}$, we should reject the null hypothesis. For *p*-value approach, if p-value<α, we should reject the null hypothesis and state the conclusion that there is at least one model has different performance at the significance level of α=0.05.

If the null hypothesis that all models have the same performance is rejected, it indicates that the performance of the models is significantly different, necessitating proceeding with a post hoc test to distinguish which models perform better than others. There are several available pathways to realize post hoc test. If we want to compare all the models with each other, the Nemenyi test [90,91] is a commonly used method. If we only aim to compare all models with a control model, such as comparing your proposed model with the state-of-the-art models, the Bonferroni correction procedure, step-up procedure, or step-down procedure are also appropriate [92,93,94,95]. Here, we take Nemenyi test as an example of post hoc test after Friedman test. A more detail description about other procedures, please refer to [96,97,98].

To implement the Nemenyi test, we first define a critical difference (CD) as Formula (Equation 32).
(32)$$CD=q_{\alpha }\sqrt{\frac{k\left(k+1\right)}{6N}}$$
where $q_{\alpha }$ (Table 5) is the critical value based on the Studentized range statistic divided by $\sqrt{2}$.

If the difference between the average rank of two models exceeds the critical difference, the assumption that the two algorithms perform the same should be rejected with corresponding confidence. For the example in Table 4, $CD_{0.05}=1.657$ and $CD_{0.10}=1.414$; $|R_{1}-R_{2}|=1.125$, $|R_{1}-R_{3}|=1.500$ and $|R_{2}-R_{3}|=0.375$. With the significant level of 0.05, there is no difference in all of the models. With the significant level of 0.05, model *A* performs better than model *B*, as the difference of their average ranks exceeds 1.414.

## 3. Liquid Biopsy Components

During the formation and growth of primary tumors, cells undergo active release, necrosis, or apoptosis [99,100]. In these process, various components are released into the liquid, including circulating tumor cells, cell-free DNA, circulating tumor DNA, cell-free RNA, exosomes, and tumor educated platelets(TEPs) [101].

### 3.1. Circulating Tumor Cells

The presence of circulating Tumor Cells (CTCs) was firstly identified by Ashworth (Australia) in 1869 [102]. When Ashworth performed an autopsy on a metastatic breast cancer patient, cells similar to those from the primary tumor were found in the blood. CTCs are currently defined as the tumor cells that shed or migrate actively into the vessel from the primary tumor or metastatic sites and then circulate in the bloodstream [103]. The opinion of tumor self-seeding suggests that CTCs can recirculate back, resulting in the possibility of metastases, which is responsible for the majority of deaths associated with cancer [104,105]. As the access to peripheral blood circulation is a prerequisite for distant metastasis of tumors [106], detection of tumor cells in blood will indicate the possibility of distant metastasis of tumors [107].

Although the content of CTCs is extremely rare, CTCs is still a potential alternative to invasive biopsies as an origin of tumor tissue for the detection and monitoring of cancers [108,109,110,111]. These circulating tumor cells can be enriched and detected via different technologies that take advantage of their physical and biological properties [112]. The technology to obtain these cells is an evolving field of research and is challenged by the ability to isolate CTCs in a condition that can be utilized for molecular analysis and propagation into CTC derived xenografts [113].

CTC is isolated from peripheral blood, which can avoid invasive and complex biopsy procedures. The culture of tumor cell lines takes a long time and is homogeneous, which cannot accurately reflect the genetic diversity and the changing tumor microenvironment. In contrast, CTCs-derived xenografts can reflect the biological characteristics of cancer more accurately, providing a visual window for studying the dynamic evolution of cancer and allowing monitoring of the longitudinal evolution of tumors at the molecular level.

As a marker of early diagnosis, CTCs also has some limitations. A reasonable and effective enrichment method is the most important and urgent problem to be solved. The main challenge is to obtain a sufficient number of CTCs that are optimally available for further evaluation. Besides, techniques for assessing the molecular characteristics of CTCs are still evolving, and the standard in this effort for clinical practice should be unified.

### 3.2. Cell-Free DNA and Circulating Tumor DNA

In 1948, Mandel and Metais, researchers from France, firstly found nucleic acids circulating in the human blood [114,115]. Circulating cfDNA refers to the DNA which is released into the blood by necrotic or apoptotic cells, or active release [116,117]. For cancer patients, part of cfDNA comes from tumor cells. This subpopulation of the cfDNA is ctDNA. In 1977, scientists firstly confirmed the presence of ctDNA in the blood of cancer patients [118]. ctDNA is single or double stranded [113] and comes from either living, dying tumor cells or CTCs [119,120,121]. The majority of cfDNA are released from normal cells. Therefore, ctDNA only occupies a small proportion of the cfDNA [101].

The concentration of cfDNA in the blood could increase owing to certain events such as cancer, autoimmune, smoking, pregnancy, intense exercise and tissue damaging therapies [122,123,124,125,126,127,128]. Likewise, the fraction of ctDNA may vary due to various factors [129]. Although ctDNA analysis provides a viable option for the diagnosis of early cancer, existing techniques cannot overcome the difficulties of sensitivity analysis. How to standardize the testing method is still a problem to be solved.

### 3.3. Cell-Free RNA

In 1993, Lee firstly discovered the miRNA [130], which are intracellular non-coding RNA molecules containing about 22 nucleotides. The miRNAs play an important signaling role by mediating the post-transcriptional silencing in various cellular activity [131]. Circulating or cell-free miRNA (cfRNA) refers to those miRNAs that identified in the biological fluids [131]. The high turnover rate of tumor cells needs the high expression of specific genes, leading to the large amounts generation of cfRNA [132]. Therefore, researchers identified the corresponding alteration in the blood of cancer patients [101,133].

The limitation of miRNA is reflected in the inconsistency in the selection of internal or external reference genes for quantitative detection; miRNAs from different sources, such as plasma, serum, whole blood, and exosomes, have differences in quality and quantity during the separation process; Some studies have small sample sizes, which may lead to unreliable results. Therefore, the isolation and quantification of miRNA and the methods used for data analysis still need more verification.

### 3.4. Exosomes

Exosomes were first discovered in sheep reticulocytes in 1983 and named by Johnstone in 1987 [134]. It refers to the vesicles released by cells, containing an abundance of proteins, genetic information such as DNA and RNA, and other analytes [135]. With a diameter between 30 nm to 100 nm, it can be detected from plasma, saliva, urine, breast milk, hydrothorax, cerebrospinal fluid, semen and other body fluids [136]. Furthermore, it is stable in extreme pH (pH = 1–13) or freeze-thaw [137]. Since playing a key role in tumor growth and metastasis, the complicated impact of exosomes in cancer mechanism needs to be further studied. These concepts support the potential of exosomes and their components to be applied in the detection of cancer [138].

Exosomes have vesicles that enhance the stability of wrapped genetic components. The similarity between circulating exosomal miRNAs and tumor-derived miRNAs enables the former one potentially useful for screening tests for cancer. In addition, other genetic components inside the exosomes will enrich relevant research on tumor genetics. From the preliminary results obtained, the prospects are very promising. However, the technology of acquiring exosomes is still under development, which is also the main reason for limiting exosome-related research.

### 3.5. Tumor Educated Platelets

Platelets (also termed thrombocytes) are the second most abundant cell types in peripheral blood, existing as circulating anucleated cell fragments. The largest platelets are about 2–3 microns in diameter [139]. More recently, platelets are implicated a central role in the local and systemic responses to tumor growth [140,141]. Confrontation of platelets with tumor cells by transferring tumor-associated biomolecules (‘education’) is an emerging research field resulting in the term of tumor-educated platelets (TEPs).

## 4. Machine Learning Algorithms and Clinical Application in Early Cancer Detection Based on Liquid Biopsy

Several machine learning algorithms are used to detect cancer based on the characteristics extracted from liquid biopsy. An overview of all relevant papers are listed in the Appendix A (Table: Summary of related publications) with the direct URL of dataset if available. This section discusses and reviews the publications of the most commonly used algorithms for early cancer detection in recent 10 years. As this systematic survey aims to report wide studies related to early cancer detection based on liquid biopsy incorporating machine learning algorithms, over 400 papers were searched using the following keywords: (liquid biopsy OR exosome OR circulating tumor cell OR circulating tumor DNA OR cell free DNA OR microRNA OR tumor educated platelet) AND (cancer OR carcinoma OR adenocarcinoma OR tumor OR malignancy OR malignant disease) AND (svm OR support vector machine). We searched four extensively used machine learning algorithms by replacing the last keyword. For each algorithm, we checked the top 100 relevant publications in recent 10 years according to the following four criteria. Figure 4 is the workflow of select publications.

The research is about liquid biopsy.The research is about cancer detection.The research utilized corresponding machine learning method.For several models compared, we only consider the model which performs best.

### 4.1. Traditional Machine Learning Algorithms

For traditional machine learning algorithms, we reviewed linear models, support vector machine and random forest.

#### 4.1.1. Linear Models

Linear models are widely used for supervised learning because of the advantage of implementation simplicity and interpretability. Linear regression, logistic regression and LASSO are some examples of linear models.

Principle of Linear Model

Given an input data {X,y} for $X=\{x_{1},x_{2},\cdots ,x_{N}\}$. Let $\hat{y}=model\left(X\right)$ denote the prediction made by a model for the given input. Coefficients (β) are parameters that define the model by assigning a coefficient to each input, and the bias or intercept is provided by an additional coefficient. The training data is used to estimate the coefficients of the logistic regression algorithm using a learning algorithm known as a maximum-likelihood estimation. The learning algorithm assumes data distribution and produces coefficients that minimize the error of probabilities of model prediction to those in the data.

The logistic regression model can be described with a matrix for the input data *X*, a vector for the output $\hat{y}$, and a vector for the coefficients β using linear algebra represented as the Formula (Equation 33).
(33)$$\hat{y}=X\cdot \beta$$

Since the above representation is identical to linear regression, which produces real values as outputs instead of class labels, a nonlinear function is used to ensure that the output of the weighted sum is a value between 0 and 1.

Logistic regression uses the logistic function, also known as the sigmoid function, to ensure class labels’ prediction. The sigmoid function is an S-shaped curve that maps a real-valued number *x* into a number between 0 and 1 using Equation (Equation 34).
(34)$$f\left(x\right)=\frac{1}{1+e^{-x}}$$

Therefore, for logistic regression, *x* in Equation (Equation 34) is replaced with the weighted sum given in Equation (Equation 35) to produce an output between 0 and 1 for two class labels 0 and 1.
(35)$$\hat{y}=\frac{1}{1+e^{-\left(X\cdot \beta \right)}}$$

The output from the model can be interpreted as a probability from a Binomial probability distribution function.

Least Absolute Shrinkage and Selection Operator (LASSO), also known as L1-norm, adds a regularization term which is used to penalize the less important features in a data by making their respective coefficient (β) zero, thereby shrinking their weights to zero. The less important features in Equation (Equation 33) having β=0 are eliminated, thereby making LASSO useful for feature selection and the creation of simple models. It is beneficial for datasets with high dimensions and high correlation. L1-norm is given by Equation (Equation 36)
(36)$$L1_{norm}=\lambda \sum_{m=0}^{p}\beta_{m}$$
where λ is the hyperparameter that controls the shrinkage. The bias of the model increases as λ increases while variance increases as λ decreases.

B.The Application of Linear Models in Early Cancer Detection

Linear models have been applied in many ways to detect several types of cancer, either recurrent or metastatic, in different parts of the body. Table 6 is an overview of relevant publications based on linear models.

Maltoni et al. [150] used a logistic regression model to evaluate the role of altered genes in breast cancer like HER2, PI3KCA for patient prognosis due to the possibility of their correlation with CF-DNA quantity. They collected serum samples from 58 non-relapsed and 21 relapsed patients and analyzed the samples for cfDNA integrity and quantity of all oncogenes. To determine the ability of these genes in predicting a relapse, the logistic regression on a two-marker combination produced an area under curve of 0.627 with a 95% confidence interval. With further clinical validity, the study speculates the potential of cfDNA detected as liquid biopsy in clinical practice.

Gene expression information of original tissues is contained in the nucleosome footprint of cfDNA. This information can be used in the prediction of response to chemotherapy. Yang et al. [149] utilized LASSO to evaluate transcription start site (TSS) regions coverage ability of genes. Based on cfDNA data of 85 healthy individuals and 85 individuals who are breast cancer patients, the coverage at the TSS regions was utilized for the classification of individuals into either having cancer or healthy. The LASSO model was repeated 100 times with a 5-fold cross-validation technique using the R package to prevent bias. A test was implemented using plasma from 30 healthy donors and 60 patients to validate the model independently. The model recorded a significant median AUC of 0.863 for the training cohort and 0.834 for the validation cohort. The model was able to avoid overfitting, as noticed in the recorded AUC. With the analysis, the use of cfDNA nucleosome footprints to predict neoadjuvant chemotherapy was highlighted and verified with the LASSO model. The study will improve personalized decision-making per patients’ treatment.

Due to the advancement of lung cancer by the time it is diagnosed, it is the deadliest cancer in the world [151]. El-Khoury et al. [148] used the bootstrap sampling method and LASSO penalization to deduce the suitable combination of protein necessary for predicting outcome to improve early detection and patients’ survival. With data comprising 93 healthy donors and 128 lung cancer patients, the level of plasma in 351 proteins was quantified, and the optimal threshold for the biomarker was selected. The validation of the panel was carried out with independent data of 49 healthy donors and 48 patients using logistic regression. With an AUC of 0.999, sensitivity of 0.992, specificity of 0.989, negative predictive value of 0.989 and positive predictive value of 0.992, lung cancer was detected irrespective of the cancer stage, making it possible to detect lung cancer earlier and aiding early treatment.

For early and accurate decisions on treatment strategies, an accurate diagnosis must be made. Therefore, it is vital to distinguish small cell lung cancer (SCLC) from non-small cell lung cancer (NSCLC). Non-small cell lung cancer can be further categorized as squamous cell carcinoma and inter alia adenocarcinoma. Raman et al. [147] collected public data containing 843 samples (small cell lung cancer = 68, squamous cell carcinoma = 351, and inter alia adenocarcinoma = 424) which were filtered based on histology. cfDNA was extracted was further extracted from plasma. Five classifiers, including random forest, support vector machine, multinomial logistic regression with ridge regularization, multinomial logistic regression with elastic net regularization, and multinomial logistic regression with lasso regularization were evaluated with the data using a leave-one-out cross-validation method. Due to the inability of some classifiers to deal with class imbalance, the authors used a random sampling method to make the number of samples in all classes equal to 68 to make the number of training samples equal to 204. Multinomial logistic regression with ridge regularization, based on iterative one-vs.-all receiver operating curve, had the best performance with a mean area under curve of 0.936. The coefficients of the logistic regression model detected that the prominent regions which differentiate non-small lung cell cancer from small lung cell cancer are located at the chromosome arm, and tumor fraction is a determinant of the prediction probability.

Cucchiara et al. [145], working with the metastatic case of EGFR-positive NSCLC reports the possibility of using the combination of liquid biopsy and radiomics to suggest management of the disease. This can be done by detecting new mutations early. Liquid biopsy is easy to perform, minimally invasive and can be done repeatedly to extract valuable information. cfDNA acquired from plasma of seven metastatic patients was analyzed using digital droplet PCR, and radiomic analysis was also done using computed tomography images. The authors were able to compare the EGFR mutation dynamics in cfDNA with the radiomic features. They used a logistic LASSO regression model to estimate the correlation between the variation in the radiomics features and the EGFR mutation status using a 27-fold Monte Carlo cross-validation method. The model implemented a feature reduction, and maximum likelihood estimation was done for the remaining features. Based on these performance analyses, an early decision can be made for treatment strategy. Although the authors found no significant relationship between the mutational status and tumor volume, there was also no discovered association between the clinical outcomes and the radiomic signatures.

Wei et al. [146] pointed out the need to have less invasive strategies for the early prognosis and detection of colorectal cancer to avoid distant metastasis. The authors extracted extracellular vesicles from plasma samples and used nanoparticle tracking analysis, transmission electron microscopy with western blotting to identify the extracellular vesicles. The samples contained 37 colorectal cancer patients, 22 colorectal adenoma patients and 42 non-cancerous control participants. It was discovered that circulating EV-miR-193a-5p can efficiently distinguish the three classes. Especially with an AUC of 0.752, it can distinguish colorectal cancer patients from the two other classes and with an AUC of 0.759, it can distinguish colorectal cancer from non-cancer. This shows circulating EV-miR-193a-5p can identify colorectal cancer than precancerous lesions. In addition, due to the importance of age factor in colorectal cancer, a logistic regression model was implemented to integrate the age with a cutoff of 55 years and circulating EV-miR-193a-5p. The integration of the age factor increased the area under curve from 0.752 to 0.775 and 0.759 to 0.795 for distinguishing colorectal cancer patients from the two other classes and colorectal cancer from non-cancer, respectively. The integration of the age factor using the model can quickly identify colorectal cancer in high-risk individuals.

Oral cancer, being one of the most frequent cancer in the world, Lin et al. [143] identified the correlation between the progression of oral squamous cell carcinoma and cfDNA. The identification of the biomarkers is essential to improve diagnosis and treatment. Plasma was extracted from 121 oral cancer patients and 50 individuals for control while ensuring that the cfDNA size distribution is similar in oral cancer patients and control donors. Analyses on the dataset revealed that the mean concentration of cfDNA in oral cancer patients was significantly higher than that of the control group. The adjusted odds ratios were determined using binary logistic regression analysis, and a confidence interval of 95% was achieved. With a statistical significance test of *p* < 0.05, the study established the relationship between cfDNA and oral cancer.

Due to the role that serum exosome plays in the development of cancer, Li et al. [144] identified protein content in serum exosome based on 30 samples. The samples included oral cancer patients with lymph node metastasis, oral cancer patients with no lymph node metastasis, and healthy controls. Oral cancer patients have a high rate of lymph node metastasis [152]. A binary logistic regression analysis was carried out to compare the use of four biomarkers (ApoA1, CXCL7, PF4V1, F13A1) and their combinations based on the area under curve. This study deduced that the four biomarkers from serum exosomes could help diagnose oral cancer-lymph node metastasis.

Due to the lack of early detection and resistance to chemotherapy, ovarian cancer is the most lethal cancer in gynecology [153,154]. Li et al. [142] performed a two-stage epigenome-wide association study to identify methylation biomarkers for epithelial ovarian cancer. The authors selected 24 cancer cases, and 24 age-frequency matched control cases for genome-wide methylation profiling, and 206 cancer cases with 205 age-frequency matched control cases. Independent *t*-test and $x^{2}$ test was used for the continuous and categorical variables, respectively. The correlation between the blood cell counts and the DNA methylation was estimated using Pearson correlation analysis. A logistic regression model was further built for the differentially methylated cpG sites in the validation stage, and it was evaluated based on the receiver operating characteristics curves. With the study, the identified set of blood-derived DNA methylation signatures and its association with epithelial ovarian cancer will serve as a tool for the early detection of ovarian cancer.

Linear models have been successfully applied to different cancer types, including breast cancer, colorectal cancer, oral cancer, lung cancer, etc. Ranging from classification to the selection of important features for further prognosis, the application of linear models as machine learning tools is important.

#### 4.1.2. Support Vector Machine

Support Vector Machine (SVM) [155] is a supervised learning method for solving data mining problems, first proposed by Cortes and Vapnik in 1995. It aims to build a decision boundary, which is known as the hyperplane, to separate different classes. The positive samples and negative samples each have the closest point to the hyperplane. SVM distinguishes different classes by maximizing the distance between these two points to the hyperplane.

Principle of SVM

If the data instances are $\left\{x_{i},y_{i}\right\}$ for i=1,2,3,⋯,N, where $x_{i}\in R^{d}$ and $y_{i}\in \left\{1,-1\right\}$. The two classes in the training data can be separated by a hyperplane *H*: $w^{T}\cdot x+b=0$. Furthermore, there are two hyperplanes $H_{1}$: $w^{T}\cdot x+b=1$ and $H_{2}$: $w^{T}\cdot x+b=-1$ parallel to *H*. The positive and negative samples, which are closest to *H*, just fall on $H_{1}$ and $H_{2}$, respectively. Such samples are support vectors. Margin is defined as the distance between $H_{1}$ and $H_{2}$ in Formula (Equation 37).
(37)$$Margin=\frac{2}{∥w∥}$$

SVM aims to learn an optimal separating hyperplane *H* to maximize the margin (minimize w), while keeping all the points correctly classified. This problem can be summarized as Formula (Equation 38).
(38)$$\begin{array}{cc} \min_{w,b}\; & \frac{{∥w∥}^{2}}{2}\\ s.t.\; & y_{i}\left(w^{T}\cdot x_{i}+b\right)\geq 1,\;\forall i \end{array}$$

For non-separable data, slack variable $\xi_{i}$ is defined to allow data samples to violate the margin or even misclassified. In Formula (Equation 39), *C* is the penalty parameter.
(39)$$\begin{array}{cc} \min_{w,b}\; & \frac{{∥w∥}^{2}}{2}+C\sum_{i=1}^{N}\xi_{i}\\ s.t.\; & y_{i}\left(w^{T}\cdot x_{i}+b\right)\geq 1,\\ & \xi_{i}\geq 0,\;\forall i \end{array}$$

When the true model of the dataset is nonlinear, we can map the input data $x\in R^{d}$ into a new high dimensional space $z\in R^{d}$ employing a nonlinear mapping $z=\Phi \left(x\right)$. After mapping, the problem can be summarized as Formula (Equation 40).
(40)$$\begin{array}{cc} \min_{w,b}\; & \frac{{∥w∥}^{2}}{2}+C\sum_{i=1}^{N}\xi_{i}\\ s.t.\; & y_{i}\left(w^{T}\cdot \Phi \left(x_{i}\right)+b\right)\geq 1,\\ & \xi_{i}\geq 0,\;\forall i \end{array}$$

To solve this problem, we need to rewrite the primal problem into its dual form.
(41)$$\begin{array}{cc} \max_{\alpha }\; & \sum_{i=1}^{N}\alpha_{i}-\frac{1}{2}\sum_{i=1}^{N}\sum_{j=1}^{N}\alpha_{i}\alpha_{j}y_{i}y_{j}\Phi {\left(x_{i}\right)}^{T}\Phi \left(x_{j}\right)\\ s.t.\; & 0\leq \alpha_{i}\leq C,\\ & \sum_{i=1}^{N}\alpha_{i}y_{i}=0,\;\forall i \end{array}$$

In Formula (Equation 41), $\alpha_{i}$ is the Lagrange Multiplier. The SVM dual problem contains the inner product of $\Phi \left(x_{i}\right)$, which is the high-dimensional feature vector. To simplify the calculation, kernel function is defined to replace the inner product as Formula (Equation 42).
(42)$$k\left(x_{i},x_{j}\right)=\Phi {\left(x_{i}\right)}^{T}\Phi \left(x_{j}\right)$$

B.The Application of SVM in Early Cancer Detection

As a traditional and popular machine learning method, SVM was widely used for early cancer detection. An overview of relevant reference to SVM is provided in Table 7.

Patrick et al. [156] reported a work of glioblastoma detection utilizing SVM with radial basis kernel. In this study, 1158 miRNAs collected from blood were analyzed. They applied SVM and filter based feature selection method to determine a suitable subset of miRNA biomarkers and achieved their best result based on 180 miRNAs with an accuracy of 81%, specificity of 79%, and sensitivity of 83%. Additionally, 52 miRNAs were significantly distinguished by unpaired Student’s *t*-test. On this basis, miR-128 and miR-342-3p stand out significantly with a *p*-value of 0.025 under correcting for multiple testing by Benjamini-Hochberg adjustment. This work revealed the possibility of miR-128, miR-342-3p and other important miRNA as biomarkers to detect glioblastom based on the analyses of 20 patients and 20 healthy individuals. It is also an instance of the effectiveness of SVM on a small sample dataset with high dimensions.

In 2015, Thomas Wurdinger’s team from the Netherlands published a study in Cancer Cell showing that mRNA from tumor-educated platelets (TEPS) is potential for diagnosis of various cancers and differentiation of cancer types [141]. This is the first time that the term of tumor-educated platelet proposed. They identified that 1453 mRNAs increased and 793 mRNAs decreased in TEPs compared with healthy platelets. Further analysis indicated that the increased TEP mRNAs were involved in biological processes such as vesicle-mediated transport and the cytoskeletal protein binding while the decreased mRNAs were involved in RNA processing and splicing. A pan-cancer classification based on SVM was implemented, distinguishing 228 patients in 6 cancers from 55 healthy individuals with 96% accuracy. Additionally, TEP mRNA profiles are also demonstrated to be effective in distinguishing the specific tumor type. Besides, they found that the platelet samples of patients possess distinct therapy-guiding markers confirmed in matching tumor tissue. In their further study [157], this team combined particle-swarm optimization (PSO) and SVM to detect non-small-cell lung cancer based on TEPs. PSO was utilized to identify the optimal biomarker panels from large amounts liquid biosources and to tune the parameter of SVM. They termed this pipeline PSO-enhanced thromboSeq. In 2019, they reevaluated the publicly available dataset in [157] and further validate the performance on a new platelet-RNA-sequencing dataset from a healthy donor (HD) and lower-grade glioma (LGG) samples [159]. In this manuscript, the authors not only provided a new dataset but also disclosed the code and state the operation of the code step by step. Heinhuis et al. [163] generalized the pipeline of PSO-enhanced thromboSeq to identify the biomarker for sarcoma on a dataset with 160 samples, achieving a diagnostic accuracy of 87% and AUC of 0.93.

Cario et al. [166] diagnosed oral cancer based on the Fourier-transform infrared (FTIR) spectra of salivary exosomes. The dataset is the whole saliva samples collected from 21 oral cancer patients and 13 healthy individuals. By analyzing the absorbance spectra, they found a number of differences between normal and cancer samples, including changes in the conformations of proteins, lipids and nucleic acids. Based on these findings, this work adopted the spectra absorbance bands between the 900 cm${}^{-1}$ and 3700 cm${}^{-1}$, the ratios and the area under the absorbance spectrum of different three certain band as the input features of classifiers. Principal component analysis–linear discriminant analysis (PCA–LDA) and SVM are included as the discrimination models. In terms of accuracy, SVM achieved a training accuracy of 100% and a cross-validation accuracy of 89%. PCA–LDA showed an accuracy of 95%.

Sunkara et al. [160] presented a centrifugal device for isolation of extracellular vesicles (EVs) from whole-blood. SVM was utilized to analyze the 8 biomarkers to detect 43 prostate-cancer patients from 30 healthy individuals. HSP90 achieved the highest sensitivity (86%), accuracy (88%), specificity (90%), and AUC (0.92) of all the test markers.

Guangzhe et al. [161] applied SVM to detect urothelial carcinoma (UC) from 65 patients with urothelial carcinoma, 58 with kidney cancer, 45 with prostate cancer, and 95 normal individuals by analyzing copy number alterations of urinary cfDNA.In this work, the random forest was first utilized to select the top 50 features. After feature selection, RF, SVM and LASSO were compared and SVM with linear kernel outperformed the other two models. The authors defined UCdetector as a combination of the 50 CNA features selected by the RF and the SVM classifier with linear kernel. UCdetector achieved the AUC of 0.959 under 10 repeats of random splitting on this dataset. Further validation on an independent dataset comprising 24 normal samples and 28 UC patients was implemented. The UCdetector distinguishes UC with an AUC of 0.888. To test the clinical sensitivity of selected 50 CNA features, the authors applied UCdetector on the 410 patients from TCGA and 90 patients from Chinese UTUC WGS data. UCdetector could accurately identify the upper tract urothelial cancers at the AUC of 0.996. Furthermore, the concordance performance of urinary cfDNA was reported to be more sensitive than the urinary sediment. This recent work recognized the top 50 important CNA features from 5000 original features and achieved satisfying performance on different datasets, even on tissue samples from TCGA. For further comment, it demonstrates the power of feature selection based on RF and the identity capacity of SVM.

Shicai et al. [162] combined SALP-seq and SVM as a pipeline to discover new cfDNA-based biomarkers for esophageal cancer. They studied the reads density of all promoters and found high reads density in normal samples and extremely low-density cancer samples on 49 genes. Of these, 34 genes are newly discovered biomarkers. The author further validated the relationship between esophageal cancer and these biomarkers on a dataset with 163 esophageal cancer samples and 11 normal samples. Moreover, 88 important regions associated with esophageal cancer were screened out from the whole genome and 54 of these, located in distal intergenic and proximal regulatory regions, were inferred to be potential diagnostic and prognostic markers for cancer. Additionally, 37 mutated genes, unique in pre-operation patients, were also discovered from a large amount of mutations in thousands of genes in pre- and post-operated esophageal cancer samples and normal samples. In this work, 103 epigenetic markers and 37 genetic markers were discovered for esophageal cancer. Finally, SVM was adopted to detect cancer samples based on 88 cancer-associated regions and achieved a high AUC of 1.0.

Zhang et al. [164] designed a DNA molecular computation platform involving SVM to analyze miRNA profiles from serum samples. They validate the performance based on clinical serum samples from 8 healthy individuals and 14 lung cancer patients with an accuracy of 86.4%.

In our recently published work [165], we proposed an Adaptive Support Vector Machine (ASVM) method by combining Shuffled Frog Leaping Algorithm and SVM for pan-cancer and subsequent tumor origin analysis. The proposed method was firstly validated on a cell-free DNA dataset with 423 sample records. We observed an improvement of AUC from 0.832 for SVM to 0.938 for ASVM. The proposed ASVM was competitive or outperformed the other six machine learning models on both the original dataset and additional two datasets.

#### 4.1.3. Random Forest

Random Forest (RF) is an ensemble machine learning approach consisting of randomly selected decision tree subsets for classification and regression. Leo Breiman introduced a random forest algorithm using bootstrapping in the random tree selection method in the early 2000s [167]. It was an enormous improvement in classification and regression machine learning accuracy. It uses the bag of random tree classifications to the ensemble and evaluates the overall classification for the given training and test data set.

Principle of Random Forest

The basic principle of the RF algorithm is the bootstrapping aggregation of randomly selected decision trees from given data observations. According to the Breiman [167] RF algorithm, it deals with classification and regression tasks using the random forest to learn. For the general RF regression estimation, Let *X* is the random input vector, where X∈R. We need to predict the response *Y* using the following Equation (Equation 43).
(43)m(X)=E[Y|X=x]

Now training sample $D_{n}=\left(\left(X_{1},Y_{1}\right),\cdots ,\left(X_{N},Y_{N}\right)\right)$ for independent input and goal data pairs of $D_{n}$ dataset that construct estimate with random tree *T*$m_{n}:T⟶R$ for *m* Function (Equation 43). Now RF consists of *M* numbers of random regression trees. The predicted estimation value ($m_{n}$) for the $j_{th}$ tree input *x* is defined as:(44)$$m_{n}\left(x;\Theta_{j}\;,\;D_{n}\right)=\sum_{i\epsilon D_{n}\left(\Theta_{j}\right)}\frac{{\|}_{X_{i}\epsilon A_{n}\left(x;\Theta_{j}\;,\;D_{n}\right)}Y_{i}}{Z_{n}\left(x;\Theta_{j}\;,\;D_{n}\right)}$$
where $D_{n}\left(\Theta_{j}\right)$ is the set of input data points for each tree and $A_{n}\left(x;\Theta_{j}\;,\;D_{n}\right)$ is the data elements of each input observation, $Z_{n}\left(x;\Theta_{j}\;,\;D_{n}\right)$ is preselected data for input tree construction from $A_{n}\left(x;\Theta_{j},\;D_{n}\right)$. Now final random forest estimation is:(45)$$m_{M,\;\;n}\left(x;\Theta_{1},\;\ldots \Theta_{M},\;{\;D}_{n}\right)=\frac{1}{M}\sum_{j=1}^{M}m_{n}\left(x;\Theta_{j}\;,\;D_{n}\right)$$

For RF supervised classification, it can classify both binary and multi-class datasets [168]. Let input vector and Y is a random class vector with class value 0,1. Now we can predict label Y from input X and $D_{n}$ dataset. Therefore, RF binary classifier can obtain from the random classification trees as:(46)$$m_{M,n}\left(x;\Theta_{1},\ldots \Theta_{M},D_{n}D_{n}\right)=\left\{\begin{array}{cc} 1\; & if\frac{1}{M}\sum_{j=1}^{M}m_{n}\left(x;\Theta_{j},D_{n}\right)>\frac{1}{2}\\ 0\; & otherwise \end{array}\right.$$

B.The application of Random Forest in Early Cancer Detection

In recent years, several studies employed RF for early cancer detection from different liquid biopsy data. An overview of relevant references is provided in Table 8.

Song CX et al. [169] applied the RF algorithm to predict lung cancer, pancreatic cancer, and HCC using cfDNA 5-Hydroxy-methyl-cytosine (5hmC) mark in blood plasmas. This study collected the whole-genome cfDNA 5hmC signatures from 49 patients with seven cancer types and eight healthy individuals for their sequence analysis using the 5hmC library. After sequence analysis, copy number variation (CNV) has estimated using PopSV 1.0.0 R package. The RF algorithm and Gaussian Mclust model applied using gene bodies and DhMRs for cancer type prediction with different cancer stages from forty HCC, pancreatic lung cancer patients, and healthy samples. RF algorithm achieved the highest accuracy, 87.5% and 92%, for two feature sets, gene bodies, and DhMRs, while Mclust prediction accuracies are 82.5% and 90%.

Cohen et al. [170] designed CancerSEEK method for early cancer detection using circulating protein biomarkers and mutation in cfDNA from multi-analyte blood test results consist of 1817 blood plasma samples with 1005 cancer patients with eight different type of cancer such as colorectum, liver, ovary, esophagus, pancreas, stomach, breast and lung cancer, and 812 healthy individuals. CancerSEEK method is usually applied for both binary and localize cancer detection from the mentioned blood test. For binary cancer detection, logistic regression (LR) classifier with 10-folds cross-validation involved using omega cfDNA score and eight protein biomarkers. CancerSEEK employed the random forest (RF) classifier with 10-folds cross-validation using omega cfDNA score, 39 protein biomarkers, and patient gender for cancer type localization. CancerSEEK achieved 70% average sensitivity for eight cancer types with 99% specificity, including the sensitivity levels of five cancer types from 69% to 98%.

Later, Nassri et al. [172] applied the binomial RF classifier for gliomas cancer detection with other types of cancer using cfDNA methylation profile from plasma samples. They achieved the highest sensitivity with an AUC value of 0.990.

Penson et al. [171] used the RF machine learning classifier for cancer type detection on tissue biopsies and then validated on two plasma ctDNA datasets. They achieved 73.8% accuracy with 5-folds cross-validation for 22 cancer types, including the highest accuracy of 95%, 87%, and 85% for uveal melanoma, glioma, and colorectal cancer, respectively. It also obtained 75% accuracy from plasma ctDNA genome analysis.

Wang et al. [176] used the RF model for gastrointestinal cancer detection using plasma cfDNA data. The gastrointestinal cancers include the gall bladder, stomach, esophagus, colon, bile duct, pancreas, liver, and rectum cancers. This study also analyzed the cfDNA profile of hepatocellular carcinoma, colorectal cancer, pancreatic cancer patients, and healthy individuals. It obtained the AUC of 0.960, 0.890, 0.910, respectively, using the RF model with 10-folds cross-validation.

Zhang et al. [173] employed the RF algorithm for feature selection and classification of early-stage lung cancer using circulating miRNA from the liquid biopsy with SMOTE oversampling technique. They achieved the highest 96.60% accuracy value (AUC = 0.996) with a maximum of 13 miRNA features. RF identified the top five circulating miRNA features for early lung cancer detection.

Peng et al. [177] applied the RF prediction model for early-stage pancreatic cancer detection of diabetic patients using blood-based plasma biomarkers. The RF model has identified the best biomarkers for early-stage pancreatic cancer patients considering the AUC measure using the leave-one-out cross-validation technique and obtained AUC values of 0.850 and 0.810 with and without the CA19-9 biomarker.

Hoshino et al. [174] employed the RF classifier to identified the biomarkers from extracellular vesicles and particle (EVP) for cancer detection. The research shows that EVP proteins are able to serve as biomarkers for early cancer detection and tumor origin detection. This study used 426 human EVP profile samples for cancer detection and achieved over 90% sensitivity and 88% specificity on both training set and test set.

### 4.2. Deep Learning

In cancer detection, traditional machine learning algorithms usually rely heavily on the representation of the selected information [178]. However, in most cases, it is difficult to give an effective feature set. In addition, manually designing features requires a lot of manpower and time in complex tasks. Therefore, deep learning came into being. When training the model, deep learning utilizes high-level features to represent low-level features, that is, to build complex concepts by combining simple concepts [179]. Since our survey focuses on commonly used algorithms based on the characteristics extracted from liquid biopsy, and the extracted features are substantially tabular data (i.e., a sample by feature matrix); therefore, here we just discuss the basic deep learning model without introducing the spatial-aware or time-aware blocks in computer vision or natural language processing. A classic case of deep learning is the multilayer perceptron (MLP, also named as a neural network (NN)).

Principle of MLP

A multilayer perceptron is a function that maps a set of input values to output values, and this function consists of many simpler functions [180]. It can be considered that each function gives a new representation of the input. Generally, an MLP consists of three different blocks, which are the input layer, hidden layer, and output layer. A 3-layer MLP architecture can be seen in Figure 5. Herein, the input layer accepts the features, that is, the experiment results from liquid biopsy. Hidden layers are between the input and output layers. Each hidden node in the hidden layer is a perceptron (with its own set of weights). Hidden layer can extract a feature pattern from the previous layer and model more complex functions [181]. Hidden layer is also called a fully-connected layer or dense layer. Output layer outputs the final prediction results (e.g., the binary description of sick or healthy).

Formally, for one layer:$$h=f\left(W^{T}x\right)$$
where *W* is the weight matrix (one column for each node). x is the input from the previous layer, h is the output to the next layer. f(a) is the activation function that is applied to each dimension to get the output. In most cases, Rectifier Linear Unit (ReLU) is utilized as the activate function in hidden layers since it is faster and easier to train with [182]. ReLU is an activation function defined as the positive part of its argument:$$f\left(x\right)=x^{+}=\max\left(0,x\right)$$
where *x* is the input to a node. ReLU can obtain sparse representation since most nodes will output zero. The activation functions for the output layer can be Softmax for both binary and multi-class classification. Softmax function is defined as:$$\sigma {\left(\vec{z}\right)}_{i}=\frac{e^{z_{i}}}{\sum_{j=1}^{K}e^{z_{j}}}$$
where *z* is the input vector. The term on the bottom of the formula is the normalization term which ensures that all the output values of the function will be sum to 1, thus constituting a valid probability distribution. When training, we can utilize Cross-Entropy Loss Function to optimize the neural network, which is formulated as:$$L=\frac{1}{N}\sum_{i}L_{i}=\frac{1}{N}\sum_{i}\sum_{c=1}^{M}-y_{ic}\log\left(p_{ic}\right)$$
where *M* is the number of class; $y_{ic}$ is the indicator variable (0 or 1), if the category is the same as the category of sample *i*, it is 1, otherwise it is 0; $p_{ic}$ is the predicted probability that the observation sample *i* belongs to the category *c*.

B.The Application of MLP in Early Cancer Detection

In the past several years, the utilization of neural networks in cancer detection can be summarized into two categories: feature engineering and classification. For feature engineering, a neural network is usually performed to remove the input’s noise and extract the most representative features that can best describe the subjects’ attributes. This step can also be called feature extraction or dimension reduction [183]. Regarding the neural network for classification, the architecture of the neural networks used in cancer detection varied in depth (shallow and deep architecture), loss function and other parameters [184]. An overview of relevant references is provided in Table 9.

In 2014, Krzysztof et al. [185] introduced Artificial Neural Networks (ANNs) to early lung cancer detection. The dataset, provided by Diagnostic and Monitoring of Tuberculosis and Illness of Lungs Ward in Kuyavia and Pomerania Centre of the Pulmonology (Bydgoszcz, Poland), includes 193 patients involving 48 features (i.e., blood test results, age, sex, etc.). The training set, validation set ant test set are randomly splitted with 97, 48, and 48 samples, respectively. Different ANNs are trained and analysed to achieve the best performance. The optimal architecture was composed of 3-layers MPL (48 input neurons, 9 neurons in hidden layer, 2 output neurons) with learning rate 0.1, epochs 17, linear function for hidden layer, and tangent function for output layer. The obtained classification accuracy is 97.91% and AUC is 0.9983. However, as the dataset is limited, we can not ascertain the high performance is on account of model generalization or the certain dataset splitting.

In 2016,Yunxiang et al. [186] developed a deep 6-layers Convolutional Neural Network (CNN) to detect circulating tumor cells from blood results. A training methodology utilizing k-means clustering was adopted to explore the most representative samples to build the classification boundary. The filter parameters, bis terms, and weights were automatically optimized by back-propagation under 0.1 learning rate setting. The experiment results show that the propsed CNN is superior to SVM on F-score. To validate the effectiveness of proposed training strategy, a comparison experiment was implemented, which indicates that the F-score of CNN increased from 91.2% to 97% with the training strategy. For SVM, the performance only reached 75.4% without the training method and increased to 78.4% after adopting it.

In 2018, Kothen-Hill et al. [186] proposed a CNN-based framework, named Kittyhawk, to distinguish the true cancer mutations from sequencing artifacts even in ultra-low variant allele frequencies (VAFs) at the level of $10^{-4}$. Kittyhawk is an 8-layer CNN with a fully-connected output layer, learning rate 0.1, momentum 0.9, and minibatch size 256. Kittyhawk initially proposed the read representation which combines the aligned genomic context, the quality scores, and the complete read sequence. The proposed method was first examined on 201,730 reads in the validation set, achieving an average performance of F1-score 0.961. Subsequently, the generalization capability was demonstrated on the independent lung cancer case with 0.92 F1-score reported.

In 2019, Ka-Chun Wong et al. [188] collected blood test records from 1817 patients to build three deep learning models to detect cancers as the front-line detector in a binary manner (i.e., cancer or normal). Since their datasets have standard and well-crafted input features, they directly adopted the deep feedforward neural networks with one hidden layer, two hidden layers, and three hidden layers (namely, DeepLearning1, DeepLearning2, and DeepLearning3, respectively) for model construction. The remaining training setting follows the default settings of WEKA. However, the performance of the deep learning methods cannot scale to full performance once the specificity level is relaxed.

## 5. Discussion

From the perspective of machine learning, we find out that even simple machine learning algorithms such as linear models can lead to a high-quality performance for liquid biopsy-based diagnosis for several common cancer types. However, there is no perfect model that performs the best on all datasets. Besides, the performance of machine learning models is diverse under different hyperparameter settings. To ensure the stability, we recommend Bayesian optimization for hyperparameter tuning after considering performance and runtime. With a hyperparameter optimization strategy, the machine learning model is adaptive to different datasets.

In addition, among all the machine learning models, the most popular and widely used are conventional algorithms. This is partly due to the barriers between biology and computer science; it is also partly due to the dataset size limitation. In the current data amount context, the traditional machine learning model such as linear models, support vector machine and random forest are still dominant in early cancer detection for their training speed and robustness on small dataset. We hope that the all-sided review of machine learning procedures and corresponding code demos presented in this survey can act as a reference guide. Definitely, advanced machine learning algorithms could also be applied for exploring latent biomarkers and the complicated relationship in order to further improve the performance. However, model generalization and complexity have to be balanced in a fair manner.

As limited with the sample size and the interpretability of deep learning models, deep learning was not popular in liquid biopsy cancer detection. From the related studies in the past several years, we can observe that, with the increased data amount from the liquid biopsy, deep learning methods are likely to outperform conventional machine learning methods. However, there are also concerns. The first concern is that deep learning is vulnerable to overfitting. Therefore, regularization, dropout, and early stopping are utilized to prevent neural networks from overfitting. Besides, the birth of batch normalization improves the model baselines and speeds up all structures [189]. Due to the variance shift conflict between dropout and batch normalization, these two methods are not recommended to be adopted simultaneously at bottlenecks except for high-dimentional data. Another concern is the black-box nature of deep learning [190]. Since the hidden layers between input and output layers are complex, it is difficult to extract the most important features and match them with the biological explanation. The explainable framework design is vital to introduce machine learning models into clinical application [191]. In general, the technique for explaining predictions can be categorized into backpropagation-based methods and perturbation-based methods [192]. The recent successes of explainable framework [191,192,193,194,195] do shed light on its promising ability. Therefore, we are still optimistic with its development in cancer detection in the future.

From the perspective of liquid biopsy components, we find out that machine learning is extensively used for single-omics analysis. However, a single type of circulating biomarker seldom fully reveals the essence of tumor occurrence. Therefore, multi-omics detection is another promising direction for early cancer detection and treatment monitoring. The exploration competence of machine learning can enable the capability to figure out the complex causal relationships between different molecular measurements. Therefore, the integration of machine learning methods and multi-omics, including genomics, epigenomics, transcriptomics, proteomics, metabolomics, and microbiomics, provides unprecedented opportunities to understand the underlying mechanism of tumor occurrence and early detection.

## 6. Conclusions

In this survey, we have presented an overview of machine learning protocols and the applications of different machine learning algorithms in the context of early cancer detection based on liquid biopsy. Additionally, we provided code demos for the aforementioned approaches in each procedure of machine learning. Based on the survey of over 400 papers, we have identified that the early cancer detection based on liquid biopsy has been tackled by different machine learning algorithms, which have been applied to multiple cancer types (e.g., pancreatic cancer, hepatocellular carcinoma, breast cancer, oral cancer, etc.) for a wide variety of component (e.g., circulating tumor cells (CTCs), cell-free DNA (cfDNA), circulating tumor DNA (ctDNA), cell-free RNA (cfRNA), exosomes, and Tumor Educated Platelets(TEPs)).

## Figures and Tables

**Figure 1 life-11-00638-f001:**
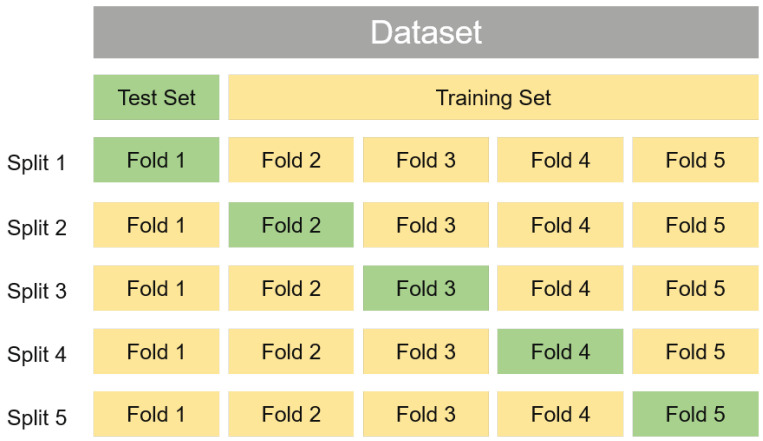
*k*-fold Cross-validation.

**Figure 2 life-11-00638-f002:**
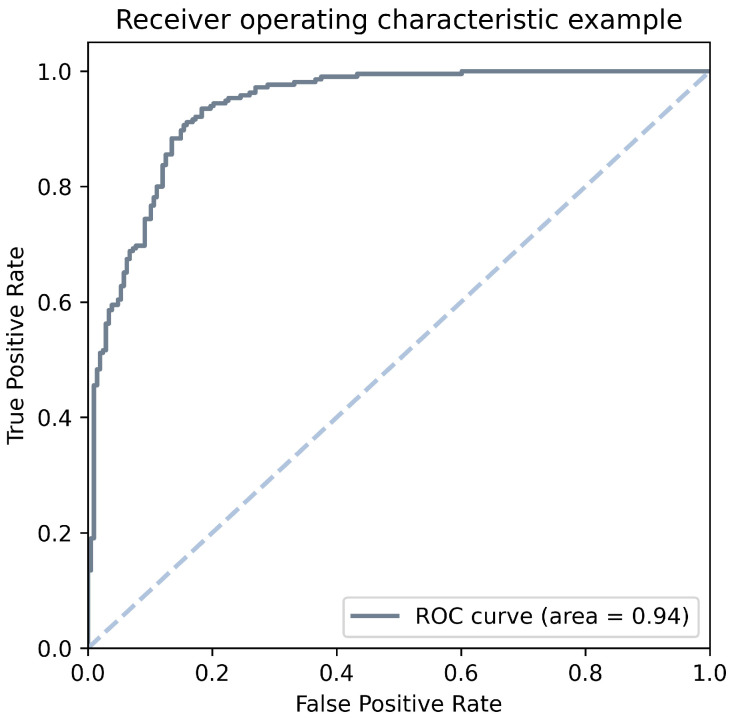
Receiver Operating Characteristic Curve.

**Figure 3 life-11-00638-f003:**
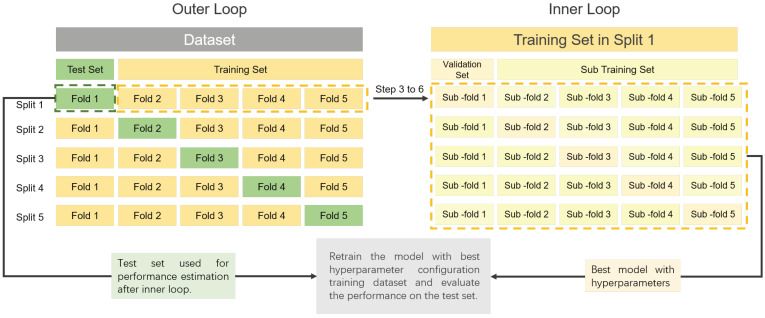
Nested 5 × 5 Cross-validation.

**Figure 4 life-11-00638-f004:**
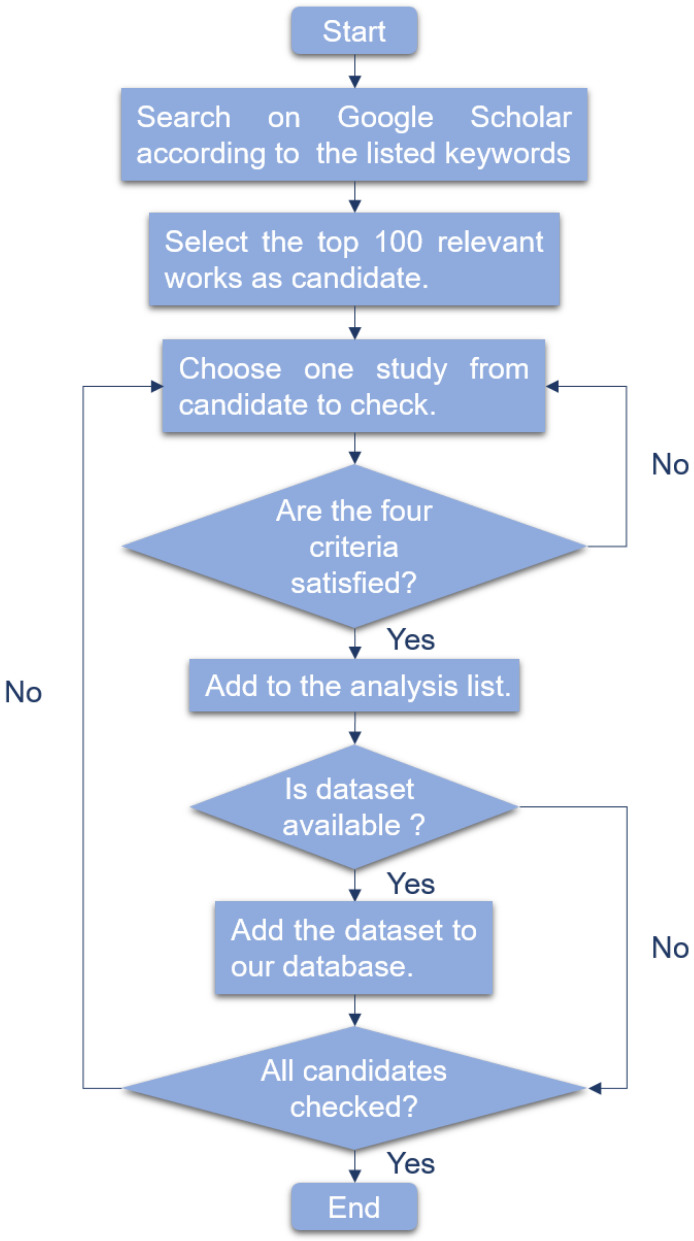
Workflow of search adn select publications.

**Figure 5 life-11-00638-f005:**
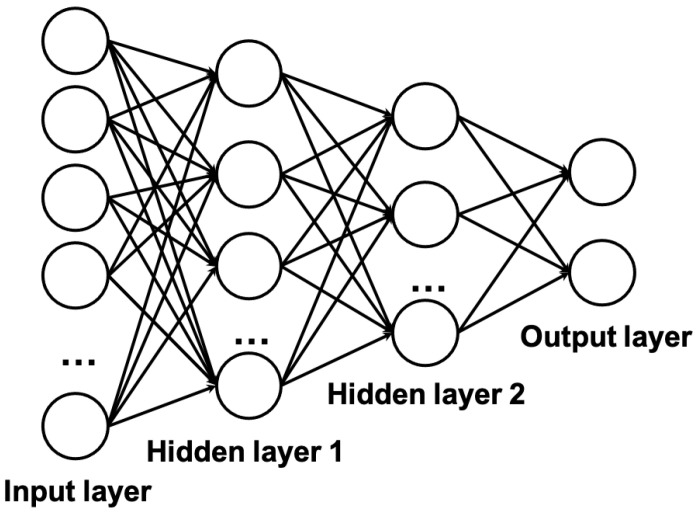
A simple 3-layer MLP architecture.

**Table 1 life-11-00638-t001:** Definition of terms.

Term	Definition
True Positive (TP)	The prediction is positive and it is actually positive.
False Positive (FP)	The prediction is positive but it is actually negative.
True Negative (TN)	The prediction is negative and it is actually negative.
False Negative (FN)	The prediction is negative but it is actually positive.

**Table 2 life-11-00638-t002:** Confusion matrix.

	Predict	Yes	No
Actual			
Yes		TP	FP
No		FP	TN

**Table 3 life-11-00638-t003:** Contingency table.

	Model *B*	Misclassification	Correct Classification
Model *A*			
Misclassification		*n* _00_	*n* _01_
Correct classification		*n* _10_	*n* _11_

**Table 4 life-11-00638-t004:** Ranks of model comparison.

Rank	Model	Model *A*	Model *B*	Model *C*
Dataset				
Dataset 1		1.5	1.5	3
Dataset 2		1	3	2
Dataset 3		1	2.5	2.5
Dataset 4		1	2	3
Average Rank		1.125	2.250	2.625

**Table 5 life-11-00638-t005:** Critical value $q_{\alpha }$ for the two-tailed Nemenyi test [97].

	*k*	2	3	4	5	6	7	8	9	10
$q_{\alpha }$										
*q* _0.05_		1.960	2.343	2.569	2.728	2.850	2.949	3.031	3.102	3.164
*q* _0.10_		1.645	2.052	2.291	2.459	2.589	2.693	2.780	2.855	2.920

**Table 6 life-11-00638-t006:** Overview of reference related to Linear Models.

Reference	Method	Dataset Availible	URL For Dataset	Cancer Type	Sample Type	Biomarker
[142]	LR	Y	https://www.ncbi.nlm.nih.gov/gds/?term=GSE31682, accessed on 29 June 2021	Ovarian	Blood	DNA methylation
[143]	LR	N		Oral cancer	Plasma	cfDNA
[144]	LR	N		Oral cancer	Blood	Exosomes
[145]	LASSO	N		Non-Small Cell Lung Cancer	Plasma	cfDNA
[146]	LR	N		Colorectal cancer	Blood	miRNA
[147]	LR, LASSO	Y	http://www.uni-koeln.de/med-fak/clcgp/, accessed on 29 June 2021	Non-small cell lung carcinoma	Blood	cfDNA
[148]	LASSO	N		Lung cancer	Plasma	Exosomes
[149]	LASSO	Y	https://identifiers.org/ncbi/insdc.sra:SRP302308, accessed on 29 June 2021	Breast cancer	Blood	cfDNA

**Table 7 life-11-00638-t007:** Overview of reference related to Support Vector Machine.

Reference	Method	Dataset Availible	URL For Dataset	Cancer Type	Sample Type	Biomarker
[156]	SVM	N		Glioblastoma	Blood	miRNA
[141]	SVM	Y	https://www.ncbi.nlm.nih.gov/geo/query/acc.cgi?acc=GSE68086, accessed on 29 June 2021	6 Cancers	Blood	TEP-RNA
[157]	PSO + SVM	Y	https://www.ncbi.nlm.nih.gov/geo/query/acc.cgi?acc=GSE89843, accessed on 29 June 2021	Non-Small-Cell Lung Cancer	Blood	TEP-RNA
[158]	SVM vs. PCA vs. LDA	N		Oral cancer	Saliva	Exosomes
[159]	PSO + SVM	Y	https://www.ncbi.nlm.nih.gov/geo/query/acc.cgi?acc=GSE107868, accessed on 29 June 2021	2 Cancers	Blood	TEP-RNA
[160]	SVM	N		prostate-cancer	Blood	Extracellular vesicles
[161]	SVM vs. RF vs. LASSO	Y	https://bigd.big.ac.cn/search/?dbId=&q=PRJCA001138, accessed on 29 June 2021	3 Cancers	Urine	cfDNA
[162]	SVM	N		Esophageal cancer	Plasma	cfDNA
[163]	PSO + SVM	N		Sarcoma	Blood	TEP-RNA
[164]	SVM	N		Lung Cancer	Serum	miRNA
[165]	SVM + SFLA vs. RF vs. KNN vs. GPC vs. GNB vs. GBM vs. SVM vs. LASSO vs. Elastic Net	Y	https://www.nature.com/articles/s41467-020-18965-w#data-availability, accessed on 29 June 2021	7 Cancers	Plasma	cfDNA

**Table 8 life-11-00638-t008:** Overview of reference related to Random Forest.

Reference	Method	Dataset Availible	URL For Dataset	Cancer Type	Sample Type	Biomarker
[169]	RF and Mclust	Y	http://www.ncbi.nlm.nih.gov/geo/query/acc.cgi?acc=GSE81314, accessed on 29 June 2021	7 cancer	Blood	cfDNA
[170]	LR and RF	Y	https://science.sciencemag.org/highwire/filestream/704651/field_highwire_adjunct_files/1/aar3247_Cohen_SM_Tables-S1-S11.xlsx, accessed on 29 June 2021	8 cancer	Blood	cfDNA and protein Biomarkers
[171]	RF	Y	https://github.com/bergerm1/GenomeDerivedDiagnosis, accessed on 29 June 2021	22 Cancers	Plasma	cfDNA
[172]	RF	Y	https://doi.org/10.5281/zenodo.3715312, accessed on 29 June 2021	intracranial tumors	Plasma	cfDNA
[173]	RF	N		Lung cancer	Serum	miRNA
[174]	RF	Y	https://www.ebi.ac.uk/pride/archive?keyword=PXD018301, accessed on 29 June 2021	5 cance	Plasma	EVP
[175]	RF	N		hepatocellular carcinoma	Blood	cfDNA
[176]	RF	N		gastrointestinal cancers	Plasma	cfDNA

**Table 9 life-11-00638-t009:** Overview of reference related to Deep Learning.

Reference	Method	Dataset Availible	URL For Dataset	Cancer Type	Sample Type	Biomarker
[185]	ANN	N		Lung cancer	Blood	Others
[186]	CNN	N		CTCs Detection	Blood	CTCs
[187]	CNN	N		Lung cancer	Blood	cfDNA
[188]	AODE, deep learning, decision tree, naive Bayes	Y	https://science.sciencemag.org/highwire/filestream/704651/field_highwire_adjunct_files/1/aar3247_Cohen_SM_Tables-S1-S11.xlsx, accessed on 29 June 2021	8 Cancers	Blood	Multianalyte

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
