# Peer review of "Machine Learning Protocols in Early Cancer Detection Based on Liquid Biopsy: A Survey"

_life, 2021, doi:10.3390/life11070638_

Round 1
Reviewer 1 Report
In this manuscript the authors reviewed machine learning protocols in early cancer detection based on liquid biopsy. This review is of interest and the overall idea is novel. There are, however, some weak points in the manuscripts which need to be improved.
Major issues
The authors mentioned the review field is machine learning in liquid biopsy. However, in Part 4 Machine Learning Algorithms and Application, the author include many studies which are not liquid biopsy. The authors should check and revise this section. For example, Ref 187.
Minor issues
Grammar should be checked. For example, Line 16, there are double “.” in the sentence “ deaths in 2020..”
Reviewer 2 Report
Major points;
- The chapters consist of Introduction, Liquid biopsy components, Machine Learning related methods, Machine learning algorithms and application and Conclusion. I would suggest that the chapters should be modified as follows;
- Introduction
- Machine learning related methods
- Deep learning related methods
- Liquid biopsy components
- Clinical application of machine learning/deep learning methods in liquid biopsy
- Machine learning
- Deep learning
- Conclusion
Indeed, the deep learning was developed from machine learning, however, the deep learning and machine learning should be discussed separately (Zhang et al. Drug Discov Today. 2017 Nov;22(11):1680-1685). There are lots of manuscript confusing the method of machine learning and deep learning. In deep learning (deep-neural network), there are certain consideration of “black-box predictions”.
- Although the manuscript is well written about the analytic methods for early cancer detection, the disadvantages of the machine learning/deep-learning and the strategy to overcome these problems also should be described. Please refer the following articles.
- The “Black-box” prediction (Loyola-Gonzalez et al. IEEE access. 2019 7: 154096-154113)
- AI early warning score (xAIEWS) system (Lauritsen et al. Nat Commun. 2020 Jul 31;11(1):3852).
- Double-descent (Nakkarian et al. ICLR. 2020; Nakkarian et al. arXiv. 2019: 1912.02292)
- Dropout/Batch normalization (Li et al. Proceedings of the IEEE/CVF International Conference on Computer Vision (ICCV). 2019: 3978-3987)
- Explainable AI/Interpretable AI (Lauritsen et al. Nat Commun. 2020 Jul 31;11(1):3852).
- Domain shift
- Domain adaptation (Choudhary et al. Yearb Med Inform. 2020 Aug;29(1):129-138.)
- Clinical application (Guidelines and legislation)
- Good machine learning practices (Artificial Intelligence/Machine Learning (AI/ML)-Based Software as a Medical Device (SaMD) Action Plan. 2021 Jan; U.S Food and Drug Administration)
- Algorithm change protocol (Artificial Intelligence/Machine Learning (AI/ML)-Based Software as a Medical Device (SaMD) Action Plan. 2021 Jan; U.S Food and Drug Administration)
- General Data Protection Regulation (Artificial Intelligence/Machine Learning (AI/ML)-Based Software as a Medical Device (SaMD) Action Plan. 2021 Jan; U.S Food and Drug Administration)
- Proposal for a Regulation laying down harmonised rules on artificial intelligence (Proposal for a Regulation laying down harmonised rules on artificial intelligence. 2021; European commission)
- The “Black-box” prediction (Loyola-Gonzalez et al. IEEE access. 2019 7: 154096-154113)
